# The role of V3 neurons in speed-dependent interlimb coordination during locomotion in mice

Han Zhang[1†], Natalia A Shevtsova[2†], Dylan Deska-Gauthier[1], Colin Mackay[1], Kimberly J Dougherty[2], Simon M Danner[2*‡], Ying Zhang[1*‡], Ilya A Rybak[2*‡]

[1]Department of Medical Neuroscience, Brain Repair Centre, Faculty of Medicine, Dalhousie University, Halifax, Canada; [2]Department of Neurobiology and Anatomy, College of Medicine, Drexel University, Philadelphia, United States

**Abstract** Speed-dependent interlimb coordination allows animals to maintain stable locomotion under different circumstances. The V3 neurons are known to be involved in interlimb coordination. We previously modeled the locomotor spinal circuitry controlling interlimb coordination (Danner et al., 2017). This model included the local V3 neurons that mediate mutual excitation between left and right rhythm generators (RGs). Here, our focus was on V3 neurons involved in ascending long propriospinal interactions (aLPNs). Using retrograde tracing, we revealed a subpopulation of lumbar V3 aLPNs with contralateral cervical projections. V3^OFF mice, in which all V3 neurons were silenced, had a significantly reduced maximal locomotor speed, were unable to move using stable trot, gallop, or bound, and predominantly used a lateral-sequence walk. To reproduce this data and understand the functional roles of V3 aLPNs, we extended our previous model by incorporating diagonal V3 aLPNs mediating inputs from each lumbar RG to the contralateral cervical RG. The extended model reproduces our experimental results and suggests that locally projecting V3 neurons, mediating left–right interactions within lumbar and cervical cords, promote left–right synchronization necessary for gallop and bound, whereas the V3 aLPNs promote synchronization between diagonal fore and hind RGs necessary for trot. The model proposes the organization of spinal circuits available for future experimental testing.

**\*For correspondence:**
smd395@drexel.edu (SMD);
ying.zhang@dal.ca (YZ);
rybak@drexel.edu (IAR)

†These authors contributed equally to this work
‡These authors shared senior authorship to this work

**Competing interest:** The authors declare that no competing interests exist.

## Editor's evaluation

This article will interest neuroscientists who study how spinal circuits control locomotion. While the role of spinal interneurons in control of left–right and flexor–extensor alternations has been studied extensively, their role in hind–forelimb coordination has not been sufficiently studied. Zhang et al. study interlimb coordination by combining experimental data and computer simulation to shed light on how a population of spinal neurons may coordinate hind and fore limbs during locomotion at different speeds.

## Introduction

Coordinated rhythmic movement of the limbs during locomotion in mammals is primarily controlled by neural circuitry within the spinal cord. This spinal circuitry includes rhythm-generating (RG) circuits (*Graham Brown, 1911*; *Graham Brown, 1914*) and multiple commissural, long propriospinal, premotor, and pattern formation neurons (*Grillner, 2006*; *Kiehn, 2006*; *Kiehn, 2011*; *Kiehn, 2016*; *Rybak et al., 2006*; *Rybak et al., 2013*; *Rybak et al., 2015*; *Jankowska, 2008*; *McCrea and Rybak, 2008*; *Danner et al., 2016*; *Danner et al., 2017*; *Danner et al., 2019*; *Ausborn et al., 2021*). It is

commonly accepted that each limb is controlled by a separate spinal RG (*Forssberg et al., 1980*; *Thibaudier et al., 2013*; *Frigon, 2017*; *Danner et al., 2019*; *Latash et al., 2020*) and that RGs controlling left and right forelimbs and left and right hindlimbs are located on the corresponding sides of cervical and lumbar enlargements of the spinal cord, respectively (*Kato, 1990*; *Ballion et al., 2001*; *Juvin et al., 2005*; *Juvin et al., 2012*). The left and right lumbar and cervical circuits are connected through multiple types of local commissural interneurons (CINs), which coordinate left–right activities (*Stein, 1976*; *Butt and Kiehn, 2003*; *Quinlan and Kiehn, 2007*; *Talpalar et al., 2013*; *Bellardita and Kiehn, 2015*; *Rybak et al., 2015*; *Shevtsova et al., 2015*). In turn, descending long propriospinal neuron (dLPNs) and ascending long propriospinal neurons (aLPNs) mediate interactions between cervical and lumbar circuits (*Juvin et al., 2005*; *Dutton et al., 2006*; *Reed et al., 2006*; *Brockett et al., 2013*; *Ruder et al., 2016*; *Flynn et al., 2017*; *Pocratsky et al., 2020*). Diverse populations of CINs and LPNs are involved in coordination of limb movements, defining locomotor gait, and controlling speed and balance during locomotion (*Danner et al., 2016*; *Danner et al., 2017*; *Kiehn, 2016*; *Ruder et al., 2016*; *Pocratsky et al., 2020*).

Experimental studies with genetic ablation, silencing, or activation of genetically identified neuron types, such as $V0_V$, $V0_D$, V1, V2a, V2b, V3, Shox2, and Hb9, allowed partial identification and/or suggestion of neuron-type-specific roles in spinal circuits and motor control, including locomotion (*Lanuza et al., 2004*; *Gosgnach et al., 2006*; *Crone et al., 2008*; *Zhang et al., 2008*; *Goulding, 2009*; *Dougherty et al., 2013*; *Talpalar et al., 2013*; *Shevtsova et al., 2015*; *Bikoff et al., 2016*; *Kiehn, 2016*; *Caldeira et al., 2017*; *Ziskind-Conhaim and Hochman, 2017*; *Dougherty and Ha, 2019*; *Falgairolle and O'Donovan, 2019*). However, so far these studies have mostly focused on lumbar circuits. Genetic identities have only begun to be ascribed to long propriospinal pathways connecting lumbar and cervical spinal segments (*Ruder et al., 2016*; *Flynn et al., 2017*). Removal of dLPNs resulted in transient periods of disordered left–right coordination in mice (*Ruder et al., 2016*); an effect that our previous computational model (*Danner et al., 2017*) attributed to excitatory diagonally projecting (commissural) $V0_V$ LPNs. Similarly, more recent experimental silencing of aLPNs was shown to affect left–right coordination in rats in certain locomotor contexts (*Pocratsky et al., 2020*); however, the specific populations involved in the effects are unknown. Although the excitatory V3 neurons are unlikely to have significant descending propriospinal projections (*Ruder et al., 2016*; *Flynn et al., 2017*), it remains unknown whether there is a subpopulation of V3 neurons that are aLPNs with involvement in fore–hind and/or left–right interlimb coordination.

In this study, we specifically focused on the potential role of V3 neurons in long propriospinal interactions between lumbar and cervical circuits controlling interlimb coordination. These neurons are defined by postmitotic expression of the transcription factor single-minded homolog 1 (Sim1). They are excitatory neurons, and the majority of them project to the contralateral side of the spinal cord (*Zhang et al., 2008*). Results from initial V3 silencing experiments suggested that V3 neurons are involved in the control of locomotion, specifically robustness of the rhythm and left–right coordination (*Zhang et al., 2008*). Our prior modeling studies suggest that V3 commissural neurons are involved in promoting left–right synchronization during synchronous gaits (*Rybak et al., 2013*; *Rybak et al., 2015*; *Shevtsova et al., 2015*; *Danner et al., 2016*; *Danner et al., 2017*) by providing mutual excitation between the extensor half-centers of the left and right lumbar RGs (*Danner et al., 2019*). Yet, these studies did not provide an explanation for the unbalanced locomotion and variable left–right coordination after inactivation of V3 neurons (*Zhang et al., 2008*). Also, these studies did not account for heterogeneity of the V3 population that was shown to contain distinct subpopulations with different biophysical properties, laminar distributions, and connectivity (*Borowska et al., 2013*; *Borowska et al., 2015*; *Blacklaws et al., 2015*; *Deska-Gauthier et al., 2020*), which may underlie different functions.

Here, we identified a subset of V3 neurons with cell bodies in the lumbar spinal cord that have direct excitatory projections to the contralateral side of the cervical enlargement. The recruitment of these neurons increased with locomotor speed. Mice with glutamatergic transmission conditionally knocked out in V3 neurons (V3$^{OFF}$ mice) had significantly reduced maximal speeds of locomotion. Moreover, even moving with relatively low and medium speeds, V3$^{OFF}$ mice lost the ability to trot stably, replacing this most typical mouse gait with a lateral-sequence walk. At higher locomotor speeds, V3$^{OFF}$ mice exhibited high step-to-step variability of left–right coordination, which could be a reason for the speed limitation observed in these animals. To determine potential connectivity and functions of

local and long propriospinal V3 neurons in spinal locomotor circuits, we updated and extended our previous computational model of spinal circuits consisting of four RGs coupled by multiple CIN and LPN pathways (*Danner et al., 2017*). Based on the novel anatomical and in vitro electrophysiological data, we included in the model V3 aLPN populations that provided diagonal RG synchronization necessary for trot, in addition to the local V3 CINs involved in left–right synchronization necessary for gallop and bound. The updated model reproduced speed-dependent gait expression in wildtype (WT) mice as well as the experimentally detected changes in the maximal speed, interlimb coordination, and gait expression in V3$^{OFF}$ mice. Taken together, our results suggest different functional roles of the local V3 CINs and V3 aLPNs in interlimb coordination and speed-dependent gait expression during locomotion.

## Results

### Experimental studies

#### Lumbar propriospinal V3 interneurons provide ascending excitatory drives to the contralateral cervical locomotor circuits

Several subpopulations of V3 neurons have been found and characterized in the mouse spinal cord (*Borowska et al., 2013*; *Chopek et al., 2018*; *Deska-Gauthier et al., 2020*). Until now, however, it has not been shown whether any V3 neurons can serve as LPNs to connect the spinal circuits in lumbar and cervical regions for fore–hindlimb coordination. Previous studies indicated that there might be only a limited number of excitatory dLPNs projecting from cervical to lumbar regions (*Ruder et al., 2016*; *Flynn et al., 2017*). Therefore, we primarily focused on studying the aLPNs with projections from lumbar to cervical regions for potential overlap with the V3 neuronal population. To do so, we injected a retrograde tracer, cholera-toxin B (CTB), into the cervical C5 to C8 region of *Sim1$^{Cre/+}$*; *Rosa26$^{floxstopTdTom}$* (Sim1TdTom) mice (*Figure 1A1*). After 7 days, we harvested the lumbar spinal cords, and then identified and mapped the tdTomato (tdTom) fluorescent protein and CTB double-positive neurons in lumbar cross sections (*Figure 1A2*). CTB and tdTom double-positive neurons were indicative of V3 neurons with ascending projections to the cervical region (V3 aLPNs). We found that V3 aLPNs were almost exclusively located contralaterally to the cervical injection sites. Overall, 30% of V3 neurons (n = 3 mice) in L1–L3 were stained by CTB but only 12% of V3 neurons in L4 to L6 were CTB positive. Even though clusters of V3 neurons were distributed across ventral to deep dorsal horn (*Zhang et al., 2008*; *Borowska et al., 2013*; *Borowska et al., 2015*; *Blacklaws et al., 2015*), the highest density of V3 aLPNs was found in deep dorsal horn, lamina IV to VI, in rostral lumbar segments (*Figure 1A3*). A relatively small number of these neurons were located in the intermediate and ventral regions, lamina VII and VIII, in caudal lumbar segments (*Figure 1A4*).

To identify lumbar V3 aLPN synapses on neurons within the cervical locomotor region, such as laminae VII/VIII and IX, we injected AAV2/9-hSyn-eGFP in the lumbar spinal cords of Sim1tdTom mice (*Figure 1B1*) and detected V3 neurons expressing GFP in the intermediate and deep dorsal regions of lumbar cord, as expected (*Figure 1—figure supplement 1*). In the cervical segments, C5–C8, we observed a broad distribution of GFP-positive axon terminals (*Figure 1B2*). We detected GFP/TdTom/Vglut2 (vesicular glutamate transporter 2) triple-positive terminals in the ventral interneuron (lamina VII/VIII) and motoneuron (lamina IX) regions (*Figure 1B2*), indicating that lumbar V3 aLPNs broadly innervate ventral cervical neurons. Interestingly, within the randomly sampled areas in these regions, tdTom-positive puncta are ~2.5% of total Vglut2/GFP puncta in the motoneuron region and ~28% in the interneuron regions (*Figure 1B2*), indicating that V3 aLPNs may strongly influence cervical RG circuits.

To test whether these ascending V3 LPNs in the lumbar segment affect motor output in the cervical region, we employed *Sim1$^{Cre/+}$*; *Ai32* (Sim1Ai32) mice, which expressed channelrhodopsin 2 (ChR2) specifically in Sim1-positive V3 neurons. Isolated spinal cords from P2–3 Sim1Ai32 mice were placed in a perfusion chamber split into two compartments. This split chamber was constructed over the thoracic T6–T8 segments with petroleum jelly (Vaseline) walls (*Figure 1C1*). Suction electrodes for electroneurogram (ENG) recordings were placed on the lumbar and cervical ventral roots. We first applied photostimulation on the cervical region, which evoked strong activation at cervical ventral roots, while the lumbar roots were silent (*Figure 1—figure supplement 2A1 and A2*). These results confirm the lack of descending V3 projections, which is consistent with previous studies (*Flynn et al.,*

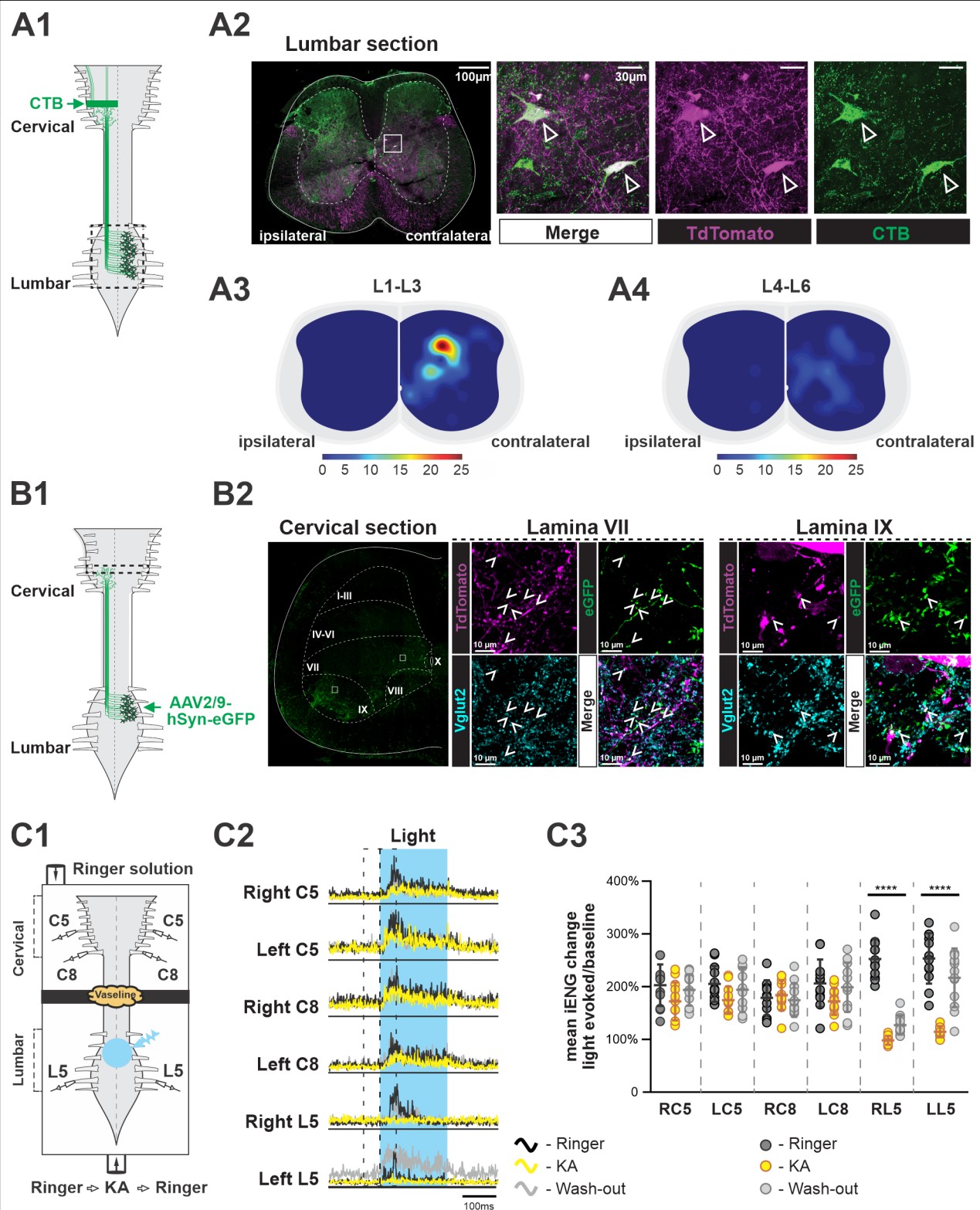

**Figure 1.** Ascending V3 long propriospinal neurons identified in the lumbar spinal cord. (**A1**) Illustration of the experimental strategy to identify lumbar V3 ascending long propriospinal interactions (aLPNs). Cholera-toxin B (CTB; green) injected into the cervical region is picked up at axon terminals and retrogradely transported to cell bodies located in the lumbar region. (**A2**) Representative image of cross section of the lumbar spinal cord of Sim1tdTom mouse (far left panel). The 'ipsilateral' and 'contralateral' sides are relative to the injection side in the cervical region. Immunohistochemical staining with different antibodies to illustrate the neurons expressing TdTom (magenta) and CTB (green) (right panels). (**A3, A4**) Color-coded heat maps of

*Figure 1 continued on next page*

*Figure 1 continued*

the distribution pattern of V3 aLPNs in the rostral (**A3**) and caudal (**A4**) lumbar segments. Scale bars of cell numbers for heat maps are shown below. (**B1**) Schematic of the experimental strategy to identify lumbar V3 aLPN synapses onto the cervical locomotor region, AAV2/9-hSyn-eGFP is injected into the lumbar region. (**B2**) Left: a representative image of a cervical hemisection injected with AAV2/9-hSyn-eGFP in the lumbar region. The dashed lines indicate approximate Rexed's laminae, and white squares indicate the approximate positions of the enlarged images. Middle (lamina VII) and right (lamina IX): representative images of the Vglut2-positive terminals (blue) of lumbar V3 aLPNs (magenta) expressing GFP (green) in cervical locomotor regions. Three random regions (~480 μm × 370 μm) in lamina IX and six regions (~370 μm × 370 μm) in lamina VII region from two cords were sampled. The Vglut2/GFP double-positive puncta (mean of 481.7 ± 47.4 in lamina IX; 143.2 ± 33.4 in lamina VII) and Vglut2/GFP/tdTom triple-positive puncta (12.7 ± 1.7 in lamina IX; 39.5 ± 9.3 in lamina VII) were quantified. (**C1**) Illustration of an isolated neonatal spinal cord placed in a split bath recording chamber. The suction electrodes for the electroneurogram (ENG) recordings were placed at ventral roots in the lumbar and cervical segments. The chamber is partitioned into two sides with a Vaseline wall (represented by yellow cloud in the figure). The photoactivation (blue light), indicated by the blue transparent circle, is on the ventral side of rostral lumbar, L1–L3, segments. (**C2**) Averaged traces of rectified ENG recordings at ventral roots of both sides of cervical (**C5, C8**) and L5 lumbar spinal segments of a P2 Sim1Ai32 mouse, before (Ringer, black), during (kynurenic acid [KA], yellow), and after washout (Washout, gray) of KA application. The time period of photostimulation is indicated by the blue shadowed area. (**C3**) The changes in integral ENG activity during light stimulation (n = 12) at the period before (dark gray circles), during (yellow circles), and after (light gray circles) KA application. Each circle represents the value of estimated parameter for one trace. Statistics for **C3** can be found in *Supplementary file 1*.

The online version of this article includes the following figure supplement(s) for figure 1:

**Figure supplement 1.** V3 neurons are infected by AAV injection in the lumbar spinal cord.

**Figure supplement 2.** Lumbar and cervical motor activity in response to photoactivation of V3 neurons in the lumbar and cervical region.

**Figure supplement 3.** Electroneurogram (ENG) recordings of cervical and lumbar roots responding to the photoactivation of lumbar V3 neurons.

*2017*; *Ruder et al., 2016*). Then, we applied photostimulation to the V3 neurons in the lumbar region. In this case, the stimulation evoked strong activity in all recorded lumbar and cervical ventral roots (*Figure 1—figure supplement 2B1 and B2*, see also *Figure 1C2 and C3*). Then, to test whether this excitation measured in the cervical roots was provided directly by ascending projections of lumbar V3 neurons, we blocked glutamatergic transmission selectively in the lumbar region with 2 mM of kynurenic acid (KA), which completely blocked NMDA and AMPA/kainate receptors (*Hägglund et al., 2010*). We found that in this case photostimulation of lumbar V3 neurons evoked small or no responses in lumbar ventral roots, while the motor responses in the cervical roots were still present (*Figure 1C2 and C3*, *Figure 1—figure supplement 3*). The lumbar responses reappeared after the drug was washed out. Such differential responses of lumbar and cervical roots to lumbar glutamatergic receptor blockade were consistent in all of our testing episodes with lumbar V3 photoactivation (*Figure 1C3*). Taken together with our anatomical findings, these results demonstrate that lumbar V3 aLPNs directly innervate the cervical spinal cord affecting cervical motor output.

## V3 aLPNs are active during locomotion

To test whether V3 aLPNs were active during locomotion, we injected CTB in the cervical region of young adult (P35–40) Sim1tdTom mice. Seven days post-injection, we subjected the animals to treadmill locomotion at either 15 cm/s or 40 cm/s for an hour. The control group was left undisturbed in the cage for one hour. Then, after one hour of rest in the home cage, which maximizes c-Fos expression in neurons (*Dai et al., 2005*), we harvested the lumbar spinal cord and used immunohistochemical staining to detect the expression of c-Fos protein in V3 neurons (*Figure 2A*). We found that the number of triple-labeled (c-Fos/CTB/tdTom) V3 neurons significantly increased after treadmill locomotion at both speeds compared to animals that only rested, but the percentage of these triple-positive V3 aLPNs almost tripled at 40 cm/s compared to 15 cm/s (*Figure 2B1–B3 and C*).

Taken together, we conclude that there are subsets of lumbo-cervical projecting commissural V3 neurons providing direct excitatory drive to cervical locomotor networks, particularly at medium locomotor speeds.

## Elimination of V3 neurons in the mouse spinal cord limits locomotor speed

To study the contribution of V3 neurons to the control of locomotion and interlimb coordination, we generated *Sim1*$^{Cre/+}$; *Slc17a6*$^{flox/flox}$ (V3$^{OFF}$) mice, in which the expression of Vglut2 was deleted in V3 neurons (*Chopek et al., 2018*). We then subjected the control (WT) and V3$^{OFF}$ mice to treadmill locomotion at different speeds. We found that the highest speed that the V3$^{OFF}$ mice could reach was around 35 cm/s (mean = 34.17 ± 6.69 cm/s; *Figure 3*), and only 3 out of 11 mice could reach 40 cm/s

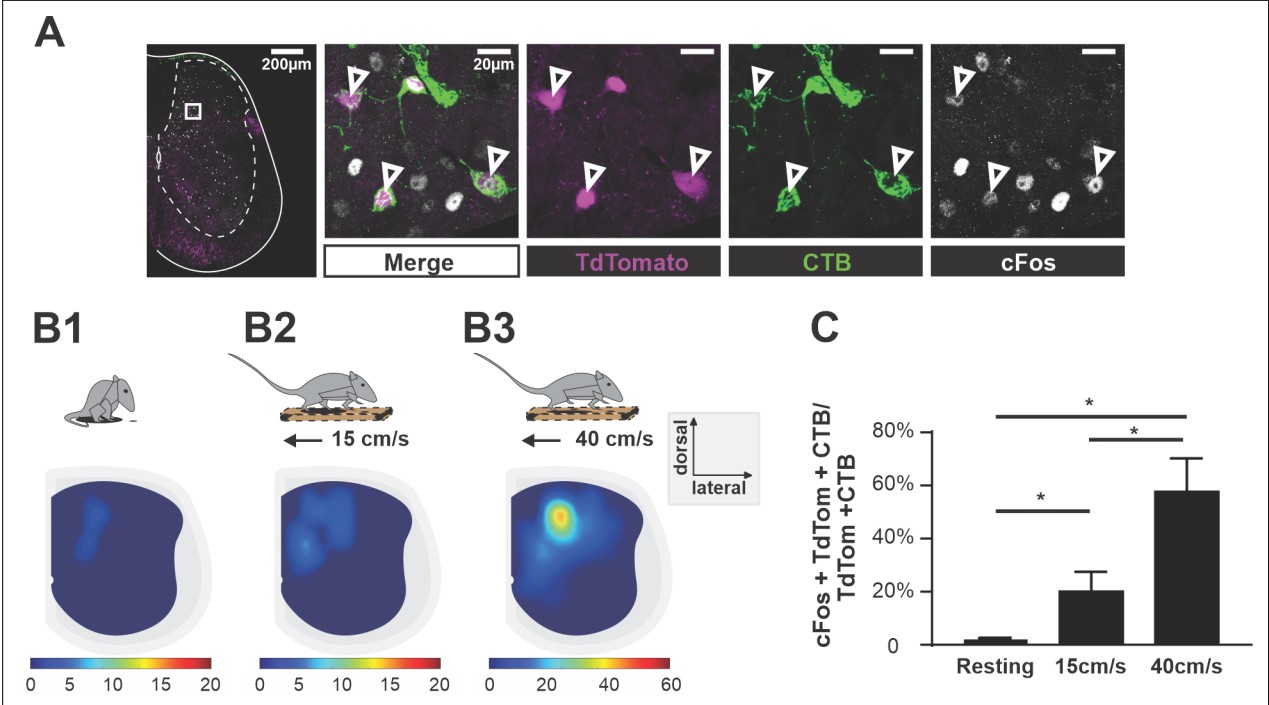

**Figure 2.** Ascending V3 long propriospinal neurons are recruited during locomotion. (**A**) Representative image of TdTom⁺ V3 neurons (magenta), cholera-toxin B (CTB) (green), and c-Fos (white) of a half cross section of the spinal cord from a *Sim1^{Cre/+}; Rosa26^{floxstopTdTom}* (Sim1tdTom) mouse. Enlarged images showing triple-positive cells. Scale bars: 200 μM and 20 μM. (**B1–B3**) Color-coded heat maps showing the distribution of c-Fos, CTB, and TdTom triple-positive V3 neurons in a cross section of lumbar spinal cord during resting (**B1**), 15 cm/s (**B2**), and 40 cm/s (**B3**) treadmill locomotion ($n_{resting}$ = 3, $n_{15cm/s}$=3, $n_{40cm/s}$ = 3). (**C**) Histogram of the corresponding percentage of triple-positive cells in TdTom/CTB double-positive population in L1–L3 segments. *$0.01<p<0.05$. Statistics for (**C**) can be found in *Supplementary file 1*.

(the maximal speed included for the subsequent analysis). Under the same experimental conditions, WT mice could run with higher speeds, up to 75 ± 7.56 cm/s (*Figure 3*). This result indicates that V3 neurons are essential for high-speed locomotion in mice.

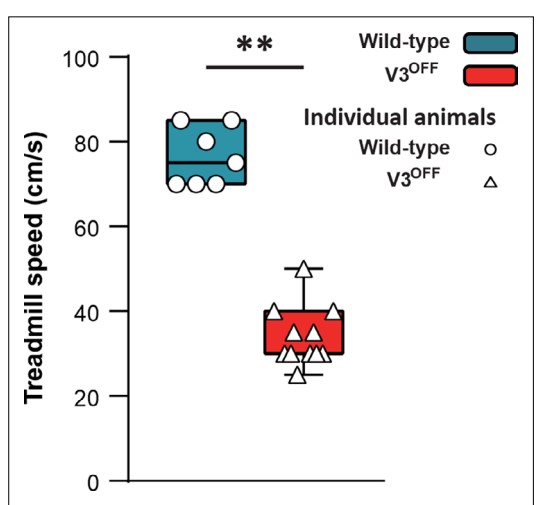

**Figure 3.** Maximum speeds of wildtype (WT) and V3^{OFF} mice on the treadmill. The box-and-whisker plots showing the highest speed of the individual WT mice (n = 7) and V3^{OFF} mice (n = 11) on the treadmill. **p<0.01. Statistics can be found in *Supplementary file 1*.

## Elimination of V3 neurons changed speed-dependent interlimb coordination

To understand the changes in interlimb coordination at low and medium speeds caused by V3 silencing, we calculated the phase differences (relative phases) between different pairs of limbs in both WT and V3^{OFF} mice: homologous (left–right), between two forelimbs and two hindlimbs; homolateral, between left forelimb and hindlimb; and diagonal, between left hindlimb and right forelimb (*Figure 4*, *Figure 4—figure supplement 1*). We then compared the corresponding relative phases and their variability between WT and V3^{OFF} mice on average (*Figure 4A1–D1*) and at different treadmill speeds (*Figure 4A2–D2*, *Figure 4—figure supplement 1*).

The left–right hindlimb and forelimb phase differences did not significantly differ between WT and V3^{OFF} mice across speeds (*Figure 4A1, B1, A2 and B2*, *Figure 4—figure supplement*

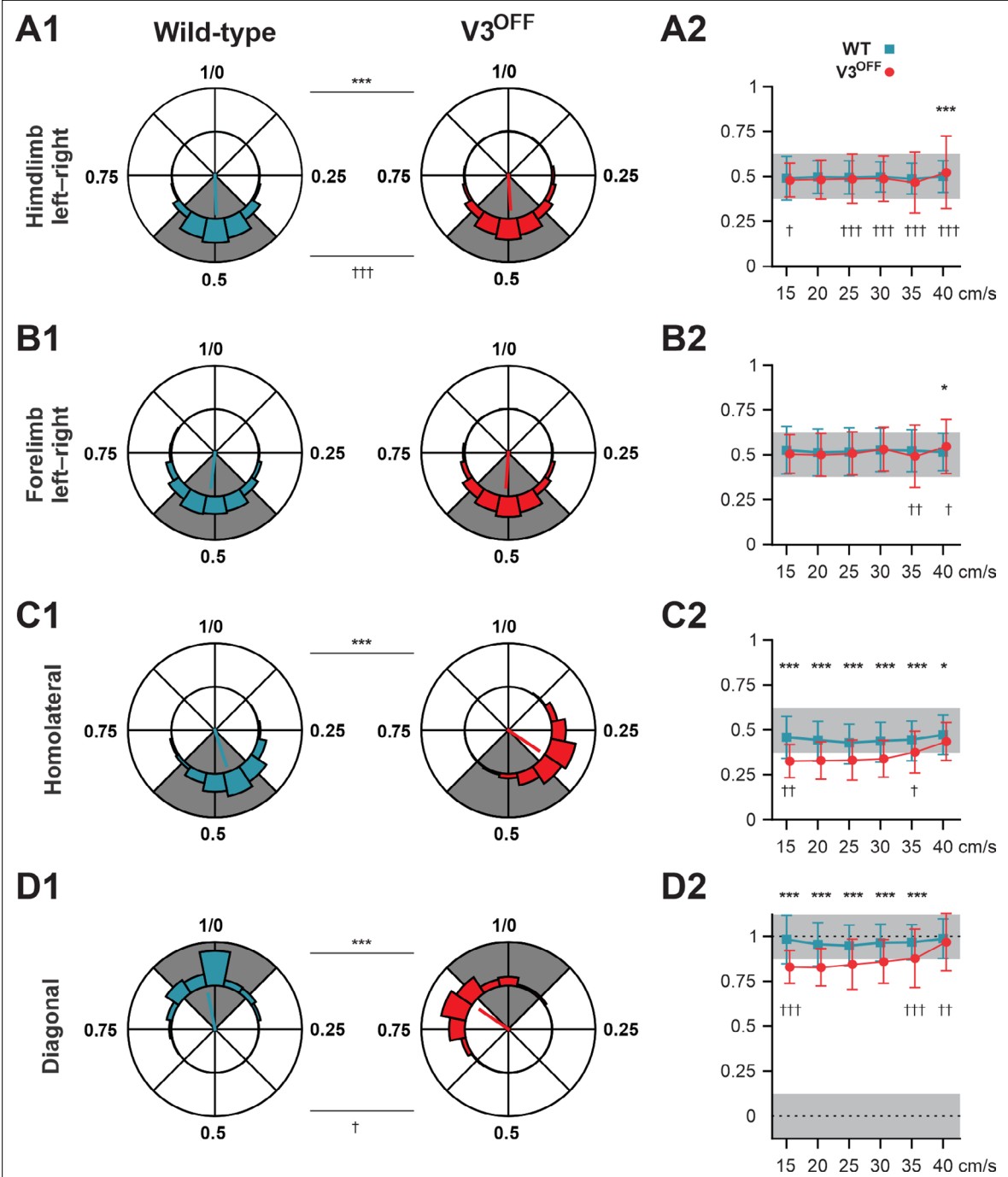

**Figure 4.** Interlimb coordination in wildtype (WT) and V3^OFF mice at different speeds. (A1–D1) Circular plots of hindlimb (A1) and forelimb (B1) left–right phase differences, homolateral phase differences (C1), and diagonal phase differences (D1) in WT (blue) and V3^OFF (red) mice. Except for the forelimb left–right phase differences, the left hindlimbs are used as the reference limb. Each vector, blue line (WT) and red line (V3^OFF), in the circular plot, indicates the mean value (direction) and robustness (radial line/length) of the phase difference. The circle is evenly separated into eight fractions. The circular histograms represent the distribution of phase differences of all steps at all tested speeds (WT, n = 1292; V3^OFF, n = 1478). (A2–D2) Plots of mean values of coupling phases at individual speeds of V3^OFF (red) and WT (blue) mice. *p<0.01; **p<0.001; ***p<0.0001 for comparisons of mean phase differences; †p<0.01; ††p<0.001; †††p<0.0001 for comparisons of the variability (concentration parameter $\kappa$) of the phase differences. Statistics for (A1–D1) and (A2–D2) can be found in *Supplementary file 1*.

The online version of this article includes the following figure supplement(s) for figure 4:

**Figure supplement 1.** Limb couplings of wildtype (WT) and V3^OFF mice at different speeds.

1A and B); mean values were close to 0.5 (perfect alternation) in both WT and V3<sup>OFF</sup> mice. The tests at individual treadmill speeds showed no significant differences, except at 40 cm/s, where the mean forelimb and hindlimb left–right phase differences of V3<sup>OFF</sup> mice differed slightly from those of the WT mice (*Figure 4—figure supplement 1A and B*). However, the variability of left–right phase differences in V3<sup>OFF</sup> mice significantly increased compared to WT mice at higher locomotor speeds (*Figure 4A2 and B2*).

In contrast to the left–right phase differences, the mean values of the homolateral (*Figure 4C1 and C2*, *Figure 4—figure supplement 1C*) and diagonal phase differences (*Figure 4D1 and D2*, *Figure 4—figure supplement 1D*) in V3<sup>OFF</sup> mice significantly deviated from those in WT mice. Post-hoc tests showed that these differences were significant across almost all tested speeds. At the highest speed (40 cm/s), the difference between the homolateral phase differences was smaller than that at lower speeds but still significant, and the diagonal phase differences did not differ significantly. This indicates that, at least at low to medium speeds, the V3<sup>OFF</sup> mice were unable to support diagonal synchronization and homolateral alternation.

## Elimination of V3 neurons changes the preferred gait from trot to walk at intermediate speeds and causes gait instability at higher speeds

Phase relationships between the four limbs define locomotor gaits (*Hildebrand, 1976*; *Hildebrand, 1980*; *Hildebrand, 1989*; *Bellardita and Kiehn, 2015*; *Lemieux et al., 2016*). Thus, the changes in the homolateral and diagonal phasing after functional removal of V3 neurons should impact gait expression.

Indeed, the step patterns and gait expression of V3<sup>OFF</sup> mice were altered in a speed-dependent manner compared to WT littermates. *Figure 5B1 and B2* shows representative stance phases of all four limbs at different treadmill speeds in WT and V3<sup>OFF</sup> mice, respectively (see also *Figure 5—videos 1–4*).

At low treadmill speeds (15 cm/s and 25 cm/s), WT mice mainly used trot (a two-beat gait characterized by diagonal synchronization and left–right alternation; *Figure 5B1*, e.g., steps $x_2$, $x_5$); however, these are transition speeds and the WT mouse exhibited a comparatively high step-to-step variability with steps that could be classified as different types of walks or even canter (a three-beat gait with only one diagonal synchronized) interspersed (e.g., steps $x_1$, $x_3$, $x_6$) between trot steps. At the same low treadmill speeds (15 cm/s and 25 cm/s), V3<sup>OFF</sup> mice used a lateral-sequence walk (a four-beat gait with longer stance than swing phases where the stance of a hindlimb is followed by that of the ipsilateral forelimb; *Figure 5B2*, e.g., steps $x_4$, $x_7$). The lateral-sequence walk of V3<sup>OFF</sup> mice was stable and consistent with little variability between the different step cycles (*Figure 5B2*, *Figure 5—video 1*, *Figure 5—video 2*).

At higher treadmill speeds (35 cm/s and 40 cm/s), the step pattern of V3<sup>OFF</sup> mice became unstable (*Figure 5B2*), while the WT mice exhibited consistent trot steps with little step-to-step variability and good synchronization of the diagonal limbs (*Figure 5B1*, e.g., steps $x_8$, $x_9$). The variable gait of the V3<sup>OFF</sup> mice (*Figure 5B2*) was characterized by steps with various left–right hindlimb and forelimb phase differences (e.g., see episodes $y_{1–3}$ in *Figure 5B2*), including in-phase steps, and a high step-to-step variability (*Figure 5—video 3*, *Figure 5—video 4*).

To further understand how V3 silencing affected speed-dependent gait expression across animals, we calculated the occurrence, persistence, and attractiveness of each gait (*Lemieux et al., 2016*) at different speeds in WT and V3<sup>OFF</sup> mice (*Figure 6*, *Figure 6—figure supplement 1*). The occurrence was the percentage of steps of a certain gait within the total steps; the persistence of a certain gait was the likelihood of a step of this gait to be followed by another step of the same gait; the attractiveness was calculated as the number of steps transferring to the target gait divided by the total number of all circumstances of gait transition in two consecutive steps.

The relative values of these parameters for each gait and the probability of transitions between different gaits at different speeds are illustrated in *Figure 6*. These analyses together clearly demonstrated that gait expression of V3<sup>OFF</sup> mice was altered compared to the WT mice in a speed-dependent manner (*Figure 6*, *Figure 6—figure supplement 1*). WT mice used trot as the preferred gait at all tested speeds (15–40 cm/s) and its occurrence, persistene, and attractiveness increased with speed (*Figure 6A*, *Figure 6—figure supplement 1A*). In contrast, the most prevalent, persistent, and attractive gait in V3<sup>OFF</sup> mice at low to medium speeds up to 35 cm/s was

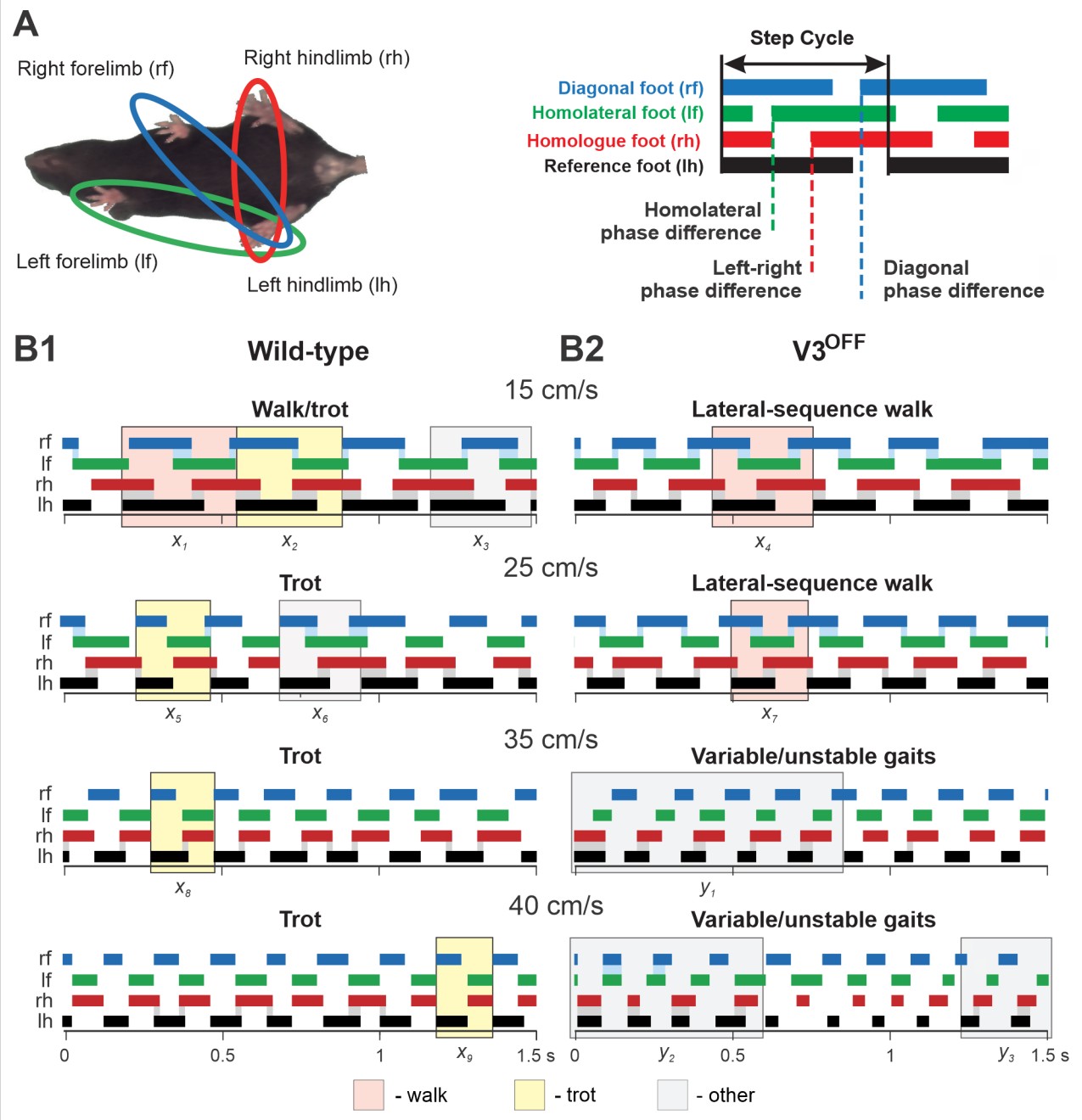

**Figure 5.** Step patterns of wildtype (WT) and V3$^{OFF}$ mice at different speeds. (**A**) Illustration of limb couplings (left). The footprint diagrams of individual limb are represented by color-coded bar graphs (right): the stance phase of each step is shown with solid bar and swing phase is the interval between two bars. The step cycle is measured from the duration between the onset of contacts of two consecutive steps of the same foot. Phase difference between the limbs was calculated as the interval between the start of stance of a specific limb and the reference limb divided by the period or step-cycle duration of the reference limb. (**B1, B2**) Representative stance phases of 1.5 s episodes of WT (**B1**) and V3$^{OFF}$ (**B2**) mice at low (15 cm/s and 25 cm/s) and medium (35 cm/s and 40 cm/s) treadmill speeds. $x_{1-9}$, exemplary steps; $y_{1-3}$, exemplary episodes referred to from the text. The shading color indicates the gait (pink, lateral-sequence walk; yellow, trot; light gray, other). Blue and gray shadows highlight periods of overlap between stance phases of the homologue limbs. The black bars show the stance phase of the reference foot (left hindlimb).

The online version of this article includes the following video for figure 5:

**Figure 5—video 1.** Wild-type and V3$^{OFF}$ mice locomotion at 15 cm/s.
https://elifesciences.org/articles/73424/figures#fig5video1

**Figure 5—video 2.** Wild-type and V3$^{OFF}$ mice locomotion at 25 cm/s.

*Figure 5 continued on next page*

a lateral-sequence walk (L-walk; *Figure 6B*, *Figure 6—figure supplement 1B*). Although V3$^{OFF}$ animals exhibited steps that could be classified as trot, the occurrence and persistence of trot episodes were low. The difference in the preferred gait between trot in WT mice and lateral-sequence or out-of-phase walk in V3$^{OFF}$ mice was associated with the changes in the homolateral and diagonal phase differences (*Figure 4C2 and D2*). As mentioned above, WT mice have trot as a dominant gait, which becomes even more prevalent and attractive with increasing speed (*Figure 6A*, *Figure 6—figure supplement 1A*). In contrast, in V3$^{OFF}$ mice the prevalence and attractiveness of the lateral-sequence walk, representing their dominant gait, decreased with speed, while the prevalence and attractiveness of less structured out-of-phase walk increased (*Figure 6B*, *Figure 6—figure supplement 1B*). This is likely associated with the increased variability of phase differences in V3$^{OFF}$ mice at higher speeds.

In summary, silencing V3 neurons limited the ability of mice to locomote at high speed (*Figure 3*), distorted interlimb coordination, and changed homolateral and diagonal phase relationships between fore- and hindlimbs at low and medium speeds (*Figure 4*). This resulted in converting gaits in V3$^{OFF}$ mice from trot to lateral-sequence walk at lower speeds and significantly reduced gait stability when speed increased (*Figures 5 and 6*).

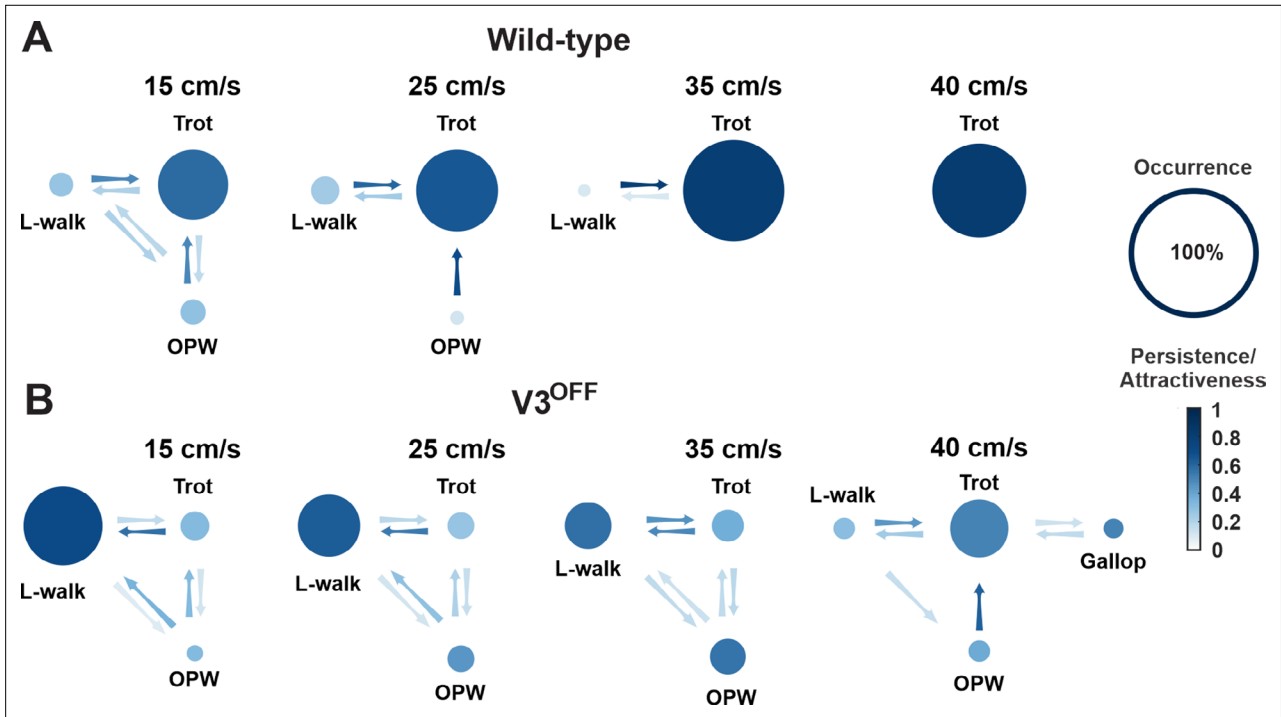

**Figure 6.** Gait transition diagrams of wildtype (WT) (**A**) and V3$^{OFF}$ (**B**) mice at different speeds. The size of circle indicates the relative occurrence of the indicated gait. The full size of the circle meaning 100% occurrence is shown on the far right. The intensity gradient of the color of the circles indicates the persistence of the corresponding gait. The intensity gradient of the color of the arrows indicates the likelihood of a transition.

The online version of this article includes the following figure supplement(s) for figure 6:

**Figure supplement 1.** Gait preference of wildtype (WT) and V3$^{OFF}$ mice at individual speeds.

## Modeling spinal locomotor circuits incorporating V3 neurons with local and long propriospinal projection

Our experimental results clearly indicate that V3 neurons should play important roles in speed-dependent interlimb coordination. Yet, the specific function and connectivity of different V3 subpopulations (i.e., V3 aLPNs and local V3 CINs) remain unknown and cannot be explicitly drawn from the current experimental data. Therefore, we extended our previous computational model of central control of interlimb coordination (*Danner et al., 2017*) and used it to study potential mechanisms by which the different V3 subpopulations may interact with the RG spinal circuitry and affect limb coordination.

The present computational model of spinal circuits was built based on our previous models (*Danner et al., 2017*; *Danner et al., 2019*). The spinal circuitry was modeled as interacting populations of neurons described in the Hodgkin–Huxley style (see 'Modeling methods'). The RG populations contained 200 neurons each; all other populations contained 100 neurons. Heterogeneity within the populations was ensured by randomizing the value for leakage reversal potential, initial conditions for the membrane potential, and channel kinetics variables among neurons. Interactions between and within populations were modelled as sparse random synaptic connections. Model equations, neuronal parameters, connection weights, and probabilities are specified in Modeling methods.

The main objectives of this study were (1) to incorporate subpopulations of V3 neurons with ascending long propriospinal projections (aLPNs) from lumbar to cervical locomotor circuits, based on our tracing and in vitro studies, and (2) to update the model so that it could reproduce multiple effects of V3 neuron removal on limb coordination, locomotor speed, and speed-dependent gait expression observed in our in vivo experimental studies.

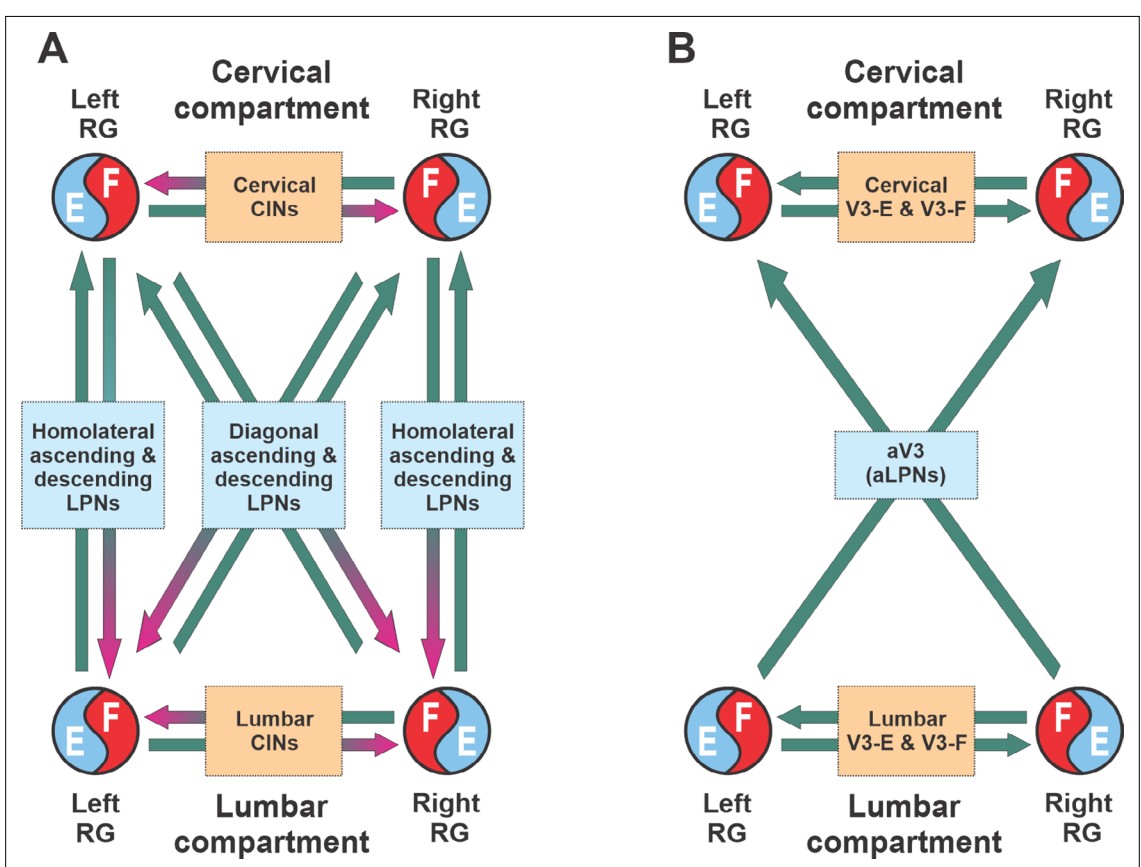

**Figure 7.** Conceptual model schematic. (**A**) The model incorporates two bilateral compartments (cervical and lumbar). Each compartment includes the left and right rhythm generators (RGs) each controlling a single limb and interacting via commissural interneurons (CINs). The cervical and lumbar compartments interact via homolaterally and diagonally projecting, descending and ascending long propriospinal neurons (LPNs). (**B**) Interactions between four RGs by different populations of V3 interneurons.

Similar to the previous models (*Danner et al., 2016*; *Danner et al., 2017*; *Ausborn et al., 2019*), our model has cervical and lumbar compartments, controlling fore- and hindlimbs, respectively (*Figure 7A*). Each compartment includes the left and right RGs, connected by several pathways mediated by local CINs. The cervical and lumbar compartments interact via homolateral and diagonal long propriospinal pathways mediated by aLPNs and dLPNs. The present model includes several distinct subpopulations of V3 neurons (*Figure 7B*). The local V3 CINs are involved in excitatory interactions between left and right extensor (V3-E) and left and right flexor (V3-F) half-centers within both cervical and lumbar compartments (*Rybak et al., 2013*; *Rybak et al., 2015*; *Shevtsova et al., 2015*; *Danner et al., 2016*; *Danner et al., 2017*; *Danner et al., 2019*). The V3 aLPNs (aV3) mediate the diagonal (commissural) pathways from lumbar to cervical RGs, which have been incorporated based on our experimental results (*Figure 1*).

## Modeling the organization of spinal circuits involved in control of locomotion

The basic architecture of the model has been developed for several years to be able to reproduce experimental data from multiple independent experimental studies (*Rybak et al., 2013*; *Rybak et al., 2015*; *Molkov et al., 2015*; *Shevtsova et al., 2015*; *Danner et al., 2016*; *Danner et al., 2017*; *Danner et al., 2019*; *Shevtsova and Rybak, 2016*; *Ausborn et al., 2019*; *Ausborn et al., 2021*; *Latash et al., 2020*) and was extended here to incorporate the results of experimental studies described above. A detailed schematic of the model is shown in *Figure 8*, and its major components are shown in *Figure 9* and described below.

### Rhythm generators

Each of the four RGs includes flexor (F) and extensor (E) half-centers, mutually inhibiting each other via inhibitory interneuron populations (InF and InE; *Figures 8 and 9*). The neurons in the F and E half-centers incorporate a slowly inactivating persistent sodium current ($I_{NaP}$) and are connected by excitatory synaptic connections allowing them to generate synchronized populational bursting activity in a certain range of an external tonic excitatory drive. As in our previous models (*Rybak et al., 2013*; *Rybak et al., 2015*; *Molkov et al., 2015*; *Shevtsova et al., 2015*; *Danner et al., 2016*; *Danner et al., 2017*; *Danner et al., 2019*; *Shevtsova and Rybak, 2016*; *Ausborn et al., 2019*; *Latash et al., 2020*), only the F half-centers operate in a bursting regime and generate intrinsically rhythmic activity, while the E half-centers receive a relatively high drive and are tonically active if uncoupled. The E half-centers generate rhythmic activity only due to inhibition from the intrinsically oscillating F half-centers. Thus, each RG generates the locomotor-like (flexor-extensor alternating) activity in a certain range of frequencies, depending on the external ('brainstem') drive to the F half-center.

### Local commissural (left–right) interactions

Left–right interactions within cervical and lumbar compartments include several commissural pathways (*Figure 9A*). Two pathways are mediated by V0 ($V0_V$ and $V0_D$) CINs and support left–right alternating activity and alternating gaits (i.e., walk and trot). The inhibitory $V0_D$ CINs provide direct mutual inhibition between the homologue flexor half-centers. The excitatory $V0_V$ CINs also provide mutual inhibition between the homologue flexor half-centers (receiving inputs from excitatory V2a and acting through inhibitory Ini1 populations). Two other commissural pathways mediated by two types of local V3 CINs (V3-E and V3-F) support synchronization of the left and right RG activities and promote left-right (quasi-) synchronized gaits (gallop and bound) at higher locomotor frequencies. The V3-F subpopulations provide mutual excitation between the F half-centers, similar to our previous models (*Rybak et al., 2015*; *Shevtsova et al., 2015*; *Danner et al., 2016*; *Danner et al., 2017*), while the V3-E CINs mediate mutual excitation between the E half-centers and are incorporated to fit our previous experimental and modeling results (*Danner et al., 2019*).

### Long propriospinal interactions between cervical and lumbar circuits

The cervical and lumbar compartments interact via descending (cervical-to-lumbar) (*Figure 9B*) and ascending (lumbar-to-cervical) LPN pathways (*Figure 9C*) whose organization is based on our previous computational models (*Danner et al., 2017*; *Ausborn et al., 2019*). The lumbar-to-cervical pathways

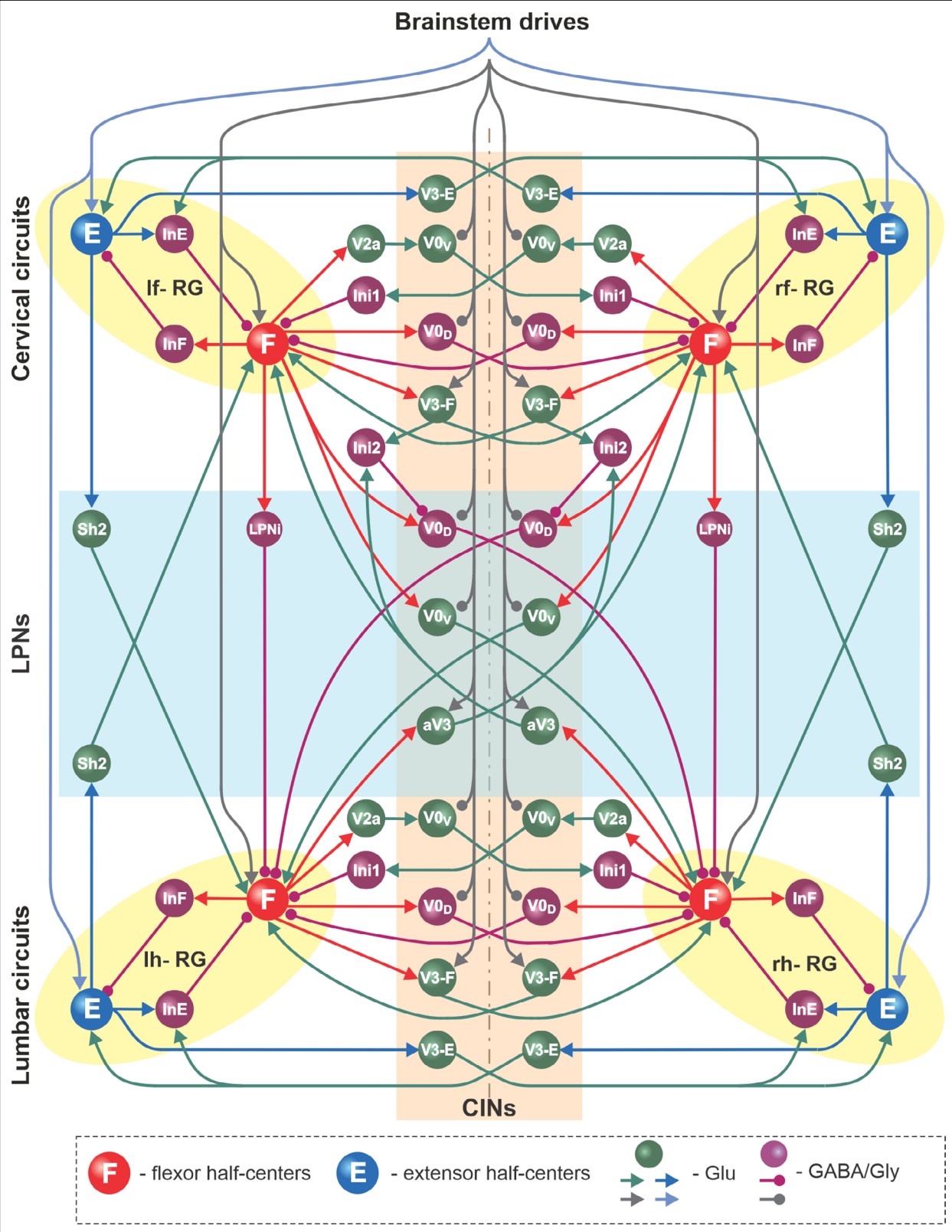

**Figure 8.** Full model schematic. Spheres represent neural populations and lines represent synaptic connections. Excitatory (Glu) and inhibitory (GABA/Gly) connections are marked by arrowheads and circles, respectively. RG, rhythm generator; CINs, commissural interneurons; LPNs, long propriospinal neurons.

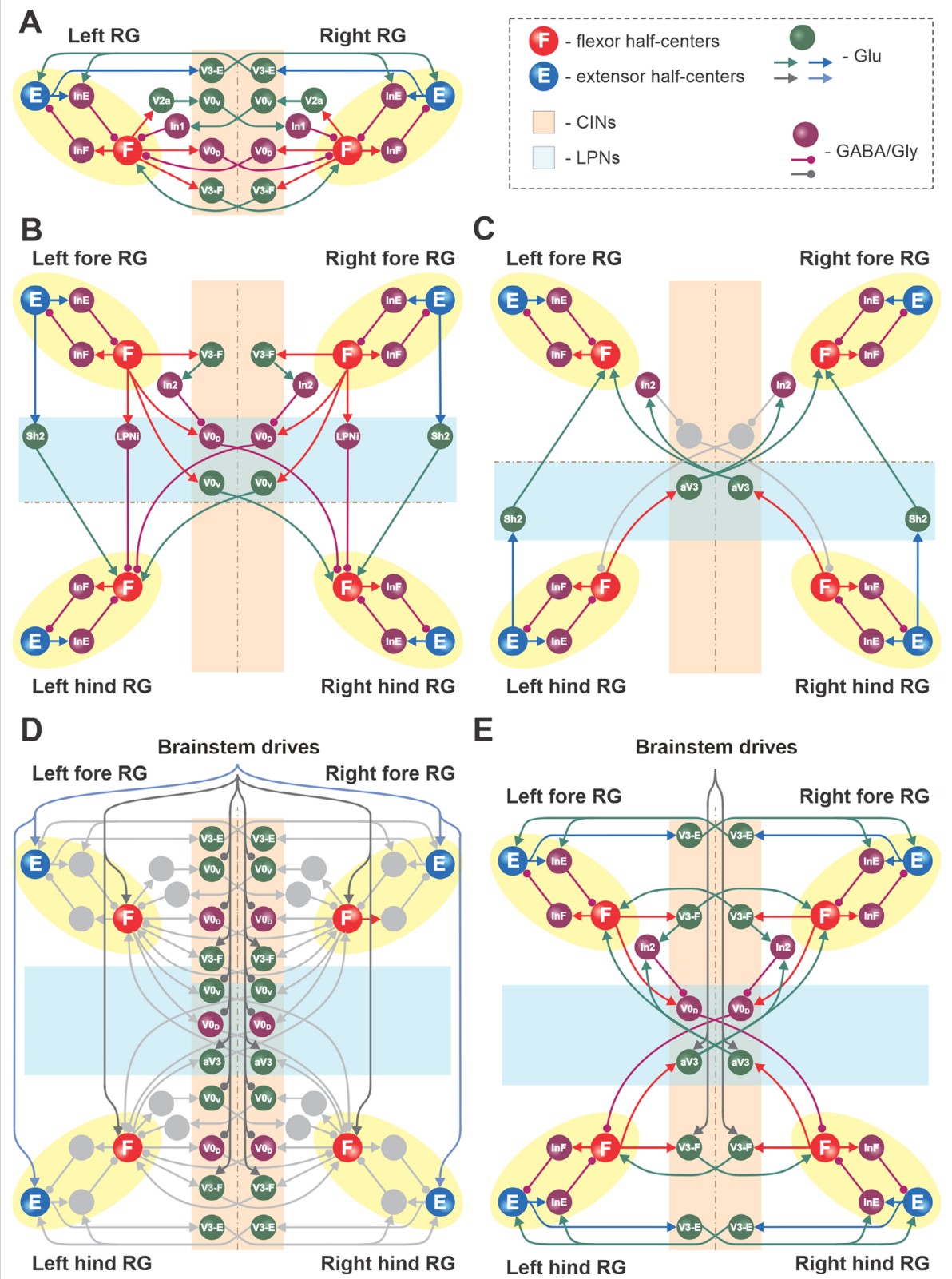

**Figure 9.** Connections within the spinal cord in the model. (**A**) Connections between the left and right rhythm generators (RGs) within each compartment (see text for details). (**B**) Cervical-to-lumbar connections via descending long propriospinal neurons (dLPNs). (**C**) Lumbar-to-cervical connections via ascending long propriospinal neurons (aLPNs). (**D**) Brainstem drive to the extensor and flexor half-centers (E and F), commissural interneurons (CINs), and long propriospinal neurons (LPNs). (**E**) Local and long propriospinal connections in the model mediated by V3 interneurons. Neural populations are shown by spheres. Excitatory and inhibitory connections between populations are shown by arrowheads and circles, respectively.

additionally incorporate ascending diagonal V3 LPNs (aV3), which is based on the present experimental data (*Figure 1*).

The descending homolateral connections (*Figure 9B*) are mediated by the excitatory Shox2 populations (Sh2), providing excitation from each cervical E half-center to its homolateral lumbar F half-center, and by the inhibitory LPN populations (LPNi), mediating inhibition of each lumbar F half-center from the homolateral cervical F half-center. The descending diagonal connections are mediated by $V0_V$ LPNs, providing excitation of each lumbar F half-center from the diagonal cervical F half-center, and by the inhibitory $V0_D$ LPNs, providing inhibition of each lumbar F half-center from the diagonal cervical F half-center. The latter pathways are also regulated by the cervical V3-F populations, which inhibit the $V0_D$ LPNs ipsilaterally through the inhibitory In2 populations.

The ascending connections in the model (*Figure 9C*) include the homolateral excitatory pathway mediated by Shox2 populations (Sh2), providing excitation from each lumbar E half-center to the homolateral cervical F half-center, and the diagonal excitatory pathway mediated by the diagonal ascending V3 (aV3) populations, providing excitation of each cervical F half-center from the diagonal lumbar F half-center. The diagonal aV3 populations also excite the In2 populations, thus regulating activity of the diagonal $V0_D$ LPNs.

## Supraspinal (brainstem) drive to cervical and lumbar circuits

The tonic brainstem drives are organized to excite all E and F half-centers, the local V3-F in both compartments, and the diagonal aV3 populations, and to inhibit the local V0 CINs in both compartments and the descending diagonal $V0_V$ and $V0_D$ LPNs (*Figure 9D*). The excitatory drive to all E half-centers was kept constant, while the value of the drive to the F half-centers, local CINs, and diagonal LPNs varied to provide control of locomotor frequency and the frequency-dependent gait expression (for details, see *Danner et al., 2016*; *Danner et al., 2017*).

## Circuit interactions mediated by V3 subpopulations

*Figure 9E* shows connectivity of all V3 subpopulations. In our previous models, we suggested that the local V3 CINs promote left–right synchronization by providing mutual excitation between left and right flexor (*Danner et al., 2016*; *Danner et al., 2017*) and left and right extensor half-centers (*Danner et al., 2019*; *Ausborn et al., 2021*). In the present model, this function is performed by V3-F and V3-E populations, respectively. Both of these populations support synchronization of left and right RG activities and promote gallop and bound at higher locomotor frequencies. Furthermore, the diagonal ascending propriospinal V3 subpopulations (aV3) mediate excitation of each cervical F half-center from the contralateral lumbar F half-center and support diagonal synchronization necessary for trot at medium locomotor frequencies. In addition, both the cervical V3-F subpopulations and the contralateral aV3 subpopulations inhibit the diagonal $V0_D$ LPNs through the inhibitory In2 populations, hence securing the stable transition from walk to trot (*Danner et al., 2017*).

## Model operation in the intact case

To show that the model with the updated V3 connectivity can reproduce speed-dependent gait expression of WT mice, we investigated the model behavior with all populations and connections intact. *Figure 10* shows the model performance for three sequentially increased values of parameter α controlling the brainstem drive to the spinal network (see *Figures 8 and 9D*). Increasing the brainstem drive caused an increase in frequency of locomotor oscillations and consecutive gait transitions from lateral-sequence walk (*Figure 10A1*) to trot (*Figure 10A2*) and finally to bound (*Figure 10A3*). *Figure 10B* shows a raster plot of activity of all neurons in the left ascending propriospinal aV3 population. When the brainstem drive was progressively increased (from $\alpha = 0$ to $\alpha = 1$), the oscillation frequency increased from 2 to 8 Hz. This increase of V3 population activity with the increase of locomotor frequency (reflecting locomotor speed) was consistent with our experimental results (*Figure 2C*).

*Figure 11A1* shows bifurcation diagrams reflecting changes in the normalized phase differences between oscillations in the left and right hind (lumbar) RGs, left and right forelimb (cervical) RGs, homolateral, and diagonal RGs with a progressive increase in brainstem drive. Increasing brainstem drive in the intact model resulted in sequential changes of gait from lateral-sequence walk to trot and then to gallop and bound (*Figure 11A1*) and was accompanied by increased frequency of stable

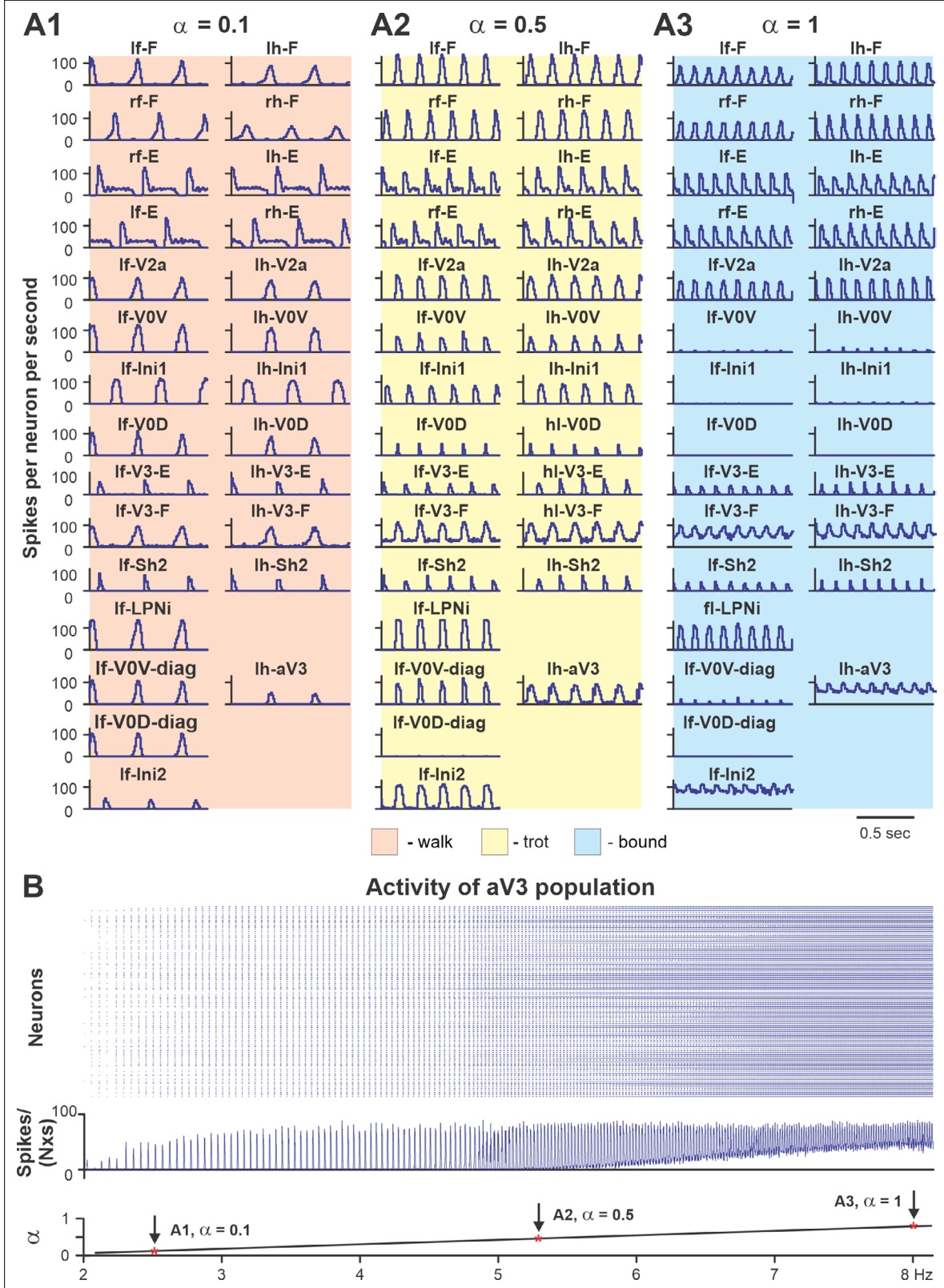

**Figure 10.** Performance of the intact model for three values of brainstem drive. (**A1–A3**) Model performance for three sequentially increased values of parameter α (α = {0.1; 0.5; 1}) that controls brainstem drive to the network. In all panels, activities of flexor and extensor (F and E) half-centers and only left cervical (fore, f-) and lumbar (hind, h-) interneuron populations are shown. Integrated activities of all populations are shown as average histograms of neuron activity [spikes/(N × s), where N is a number of neurons in population; bin = 10 ms]. (**B**) Raster plot (upper panel) of activity of the left lumbar aV3

*Figure 10 continued on next page*

*Figure 10 continued*

population when the brainstem drive (lower panel) was increased from 0.1 to 1. Note increased recruitment of aV3 neurons with the increased drive. rf, right fore; lf, left fore; rh, right hind; lh, left hind.

locomotor oscillations (*Figure 11B1*). The sequential gait changes in the model were consistent with the previous experimental observations (*Bellardita and Kiehn, 2015*; *Lemieux et al., 2016*) and our modeling studies (*Danner et al., 2016*; *Danner et al., 2017*; *Ausborn et al., 2019*). During transitions from walk to trot and from trot to gallop and bound, the model exhibited bi- or multistable behaviors (*Figure 11A1 and B1*). In these areas, a small disturbance or noise could result in spontaneous transitions of model activity from one steady state regime to the other. While building the bifurcation diagrams in *Figure 11*, noise was added to all tonic drives in the model, and, as a result, in the range of the parameter $\alpha$ from 0.4 to 0.5, spontaneous transitions between walk and trot occurred; in the range of $\alpha$ between 0.7 and 0.85, the model exhibited either trot, gallop, or bound; and at $\alpha$ values larger than 0.85 both gallop and bound coexisted. These model behaviors correspond to previous experimental and modeling studies (*Bellardita and Kiehn, 2015*; *Lemieux et al., 2016*; *Danner et al., 2017*). Thus, the proposed model organization (including V3 CIN and aLPN populations and their connectivity) can account for the speed-dependent gait expression observed in WT mice.

## Modeling the effects of elimination of V3 neurons

The model described above was used to investigate the effects of removing the V3 population to simulate our experimental studies using V3$^{OFF}$ mice, in which all V3 neurons were genetically silenced. Deletion of all V3 neurons produced notable changes in model operation (*Figure 11A2 and B2*). First, the frequency range of stable locomotor oscillation was reduced, so that above the frequency of 3.8 Hz ($\alpha$ = 0.57), the model transitioned to uncoordinated locomotor oscillations resulting in unstable, variable gaits (*Figure 11A2*). Second, the range of brainstem drive values and the range of frequencies in which the lateral-sequence walk occurred was extended, while trot and other gaits (gallop/bound) became unstable or even failed to occur (*Figure 11A2*). Both of these effects were qualitatively consistent with our experimental results (*Figures 3, 4 and 6*).

To investigate the specific role of the ascending long propriospinal V3 (aV3) neurons, we simulated a separate case when only V3 aLPN subpopulations were removed (*Figure 11A3 and B3*). Our simulations show that the deletion of only aV3 neurons in the model also leads to elimination of trot at medium locomotor frequencies. However, with further increases in brainstem drive, after some period of instability, sharp transitions to the (quasi-)synchronized gaits (gallop and bound) took place, which in this case were supported by the remaining local populations of V3-F and V3-E CINs. This suggests that V3 aLPNs support trot by promoting diagonal synchronization through excitation of the diagonal forelimb flexor half-centers by the hindlimb flexor half-centers (*Figures 8 and 9C*). This result can be considered as a modeling prediction for future experimental studies.

The effects of removing all V3 or only aV3 neurons are also illustrated in circular plot diagrams of phase differences between the activity of reference population (left hind E) and the activities of other RG (rh-E, lf-E, rf-E, lh-F, lf-F, rh-F, and rF-F) and left V3 (lh-V3-E, lh-V3-F, and lh-aV3) populations for the intact model and after deletion of all V3 or only aV3 neurons at three values of α = {0.3; 0.6; 0.9} (*Figure 11—figure supplement 1*). Each circular plot diagram shows phase differences for all steps generated for 5 s of simulation time. At $\alpha$ = 0.3, all three model versions demonstrated stable lateral-sequence walk. At $\alpha$ = 0.6, only the intact model exhibited trot while the V3- and aV3-deleted model versions had unstable variable gaits. However, at $\alpha$ = 0.9, both intact and aV3-deleted models transitioned to left–right synchronized gaits (gallop/bound) while the V3-deleted model was still unstable.

In addition, we examined our simulation results to determine the effects of removal V3 populations on gait expression for three values of brainstem drive defining locomotor frequency. The intact model exhibits a lateral-sequence walk at low locomotor frequencies and trot at medium frequencies (*Figure 12A*). After V3 removal, stable trot disappears, the model only exhibits a lateral-sequence walk at low to medium frequencies, and the interlimb coordination becomes unstable at higher frequencies, resulting in variable gaits (*Figure 12B*). The behavior of the intact model and the changes following the removal of all V3 neurons qualitatively correspond to our experimental observations (*Figures 5 and 6*).

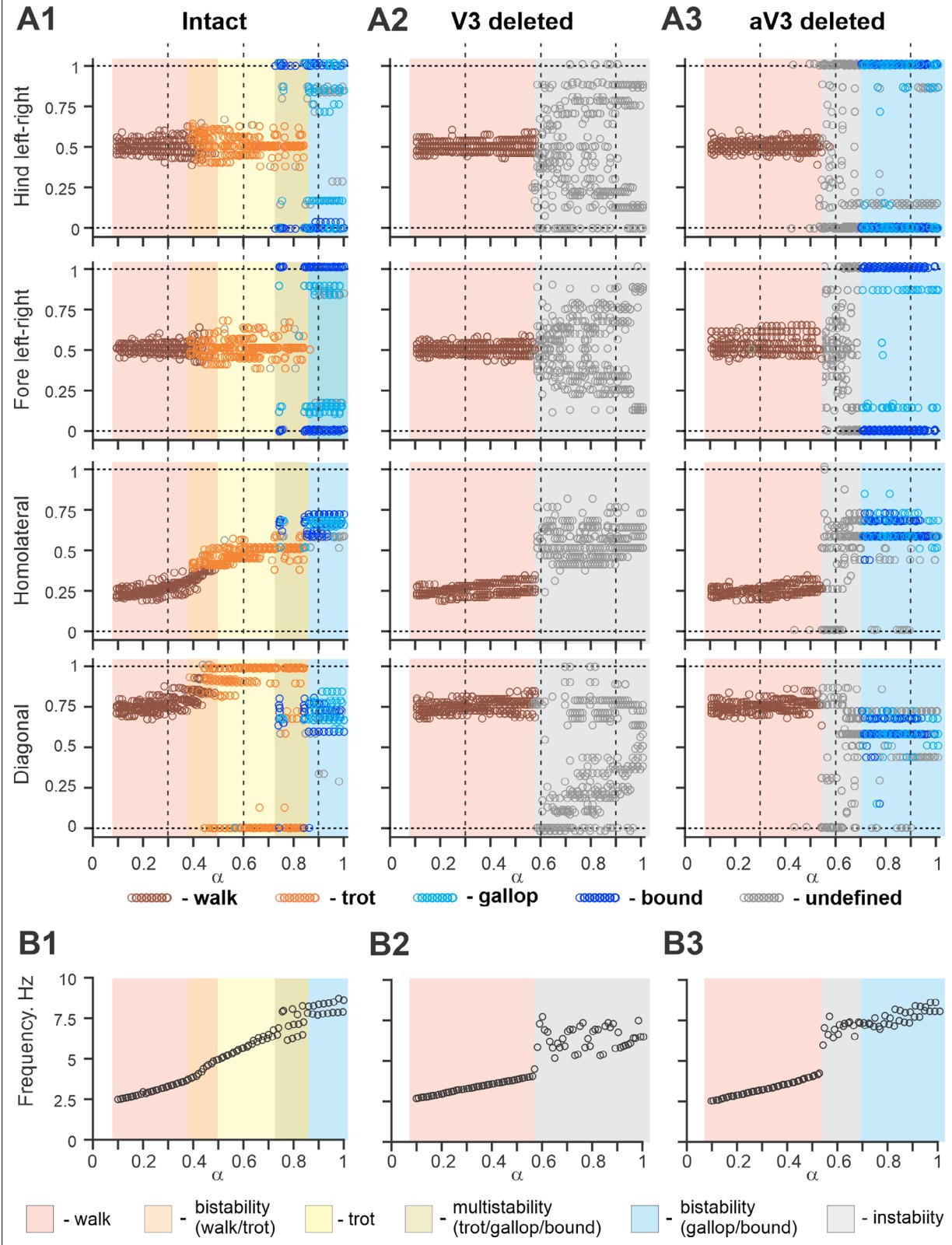

**Figure 11.** Bifurcation diagrams of the intact model, and after removal of all V3 or only propriospinal aV3 neurons. Bifurcation diagrams reflecting changes in the normalized phase differences between oscillations in the left and right hind (lumbar) rhythm generators (RGs), left and right fore (cervical) RGs, homolateral, and diagonal RGs with a progressive increase in brainstem drive ($\alpha$) from 0.1 to 1 with a step of 0.01 (see for details 'Modeling methods' and *Danner et al., 2017*). (**A1**) Intact model. (**A2**) The result of removal of all V3 neurons (V3$^{OFF}$ simulation). (**A3**) Only the diagonal aV3 are

*Figure 11 continued on next page*

removed. Parameter α that represents the brainstem drive was used as bifurcation parameters. Each diagram shows phase differences calculated for five locomotor periods when α was fixed. The vertical dashed lines correspond to three values of α = {0.3; 0.6; 0.9} for which complimentary circular plot diagrams are built showing phase differences between activities of the left hind E population (lh-E) and other relative populations for the intact model and after deletion of all V3 or aV3 neurons. (**B1–B3**) Average oscillation frequency corresponding to simulations shown in (**A1–A3**).

The online version of this article includes the following figure supplement(s) for figure 11:

**Figure supplement 1.** Phase differences for intact, V3- and aV3-deleted model for three different values of α.

A comparison of normalized left–right, homolateral, and diagonal phase differences in the intact model and after deletion of all V3 neurons (*Figure 13*) demonstrated that the intact model exhibited phase relationships typical for trot. In contrast, after removal of all V3 neurons, homolateral and diagonal phase differences were shifted towards lower values, corresponding to a lateral-sequence walk, which was similar to our experimental results (*Figure 4A2–D2*).

## Discussion

In this study, we identified a population of ascending V3 LPNs (V3 aLPNs) located in the lumbar cord that innervates and activates neurons in cervical locomotor circuits and shows increasing populational activity with increasing locomotor speed. In mice in which the entire V3 population was silenced (V3^OFF mice), the maximal locomotor speed was significantly reduced, and these mice were unable to stably trot because of a lack of synchronization between diagonal pairs of limbs. Furthermore, V3^OFF mice exhibited high step-to-step variability in left–right coordination close to their maximum locomotor speed. We extended our previous spinal circuit model (*Danner et al., 2017*) to investigate the specific

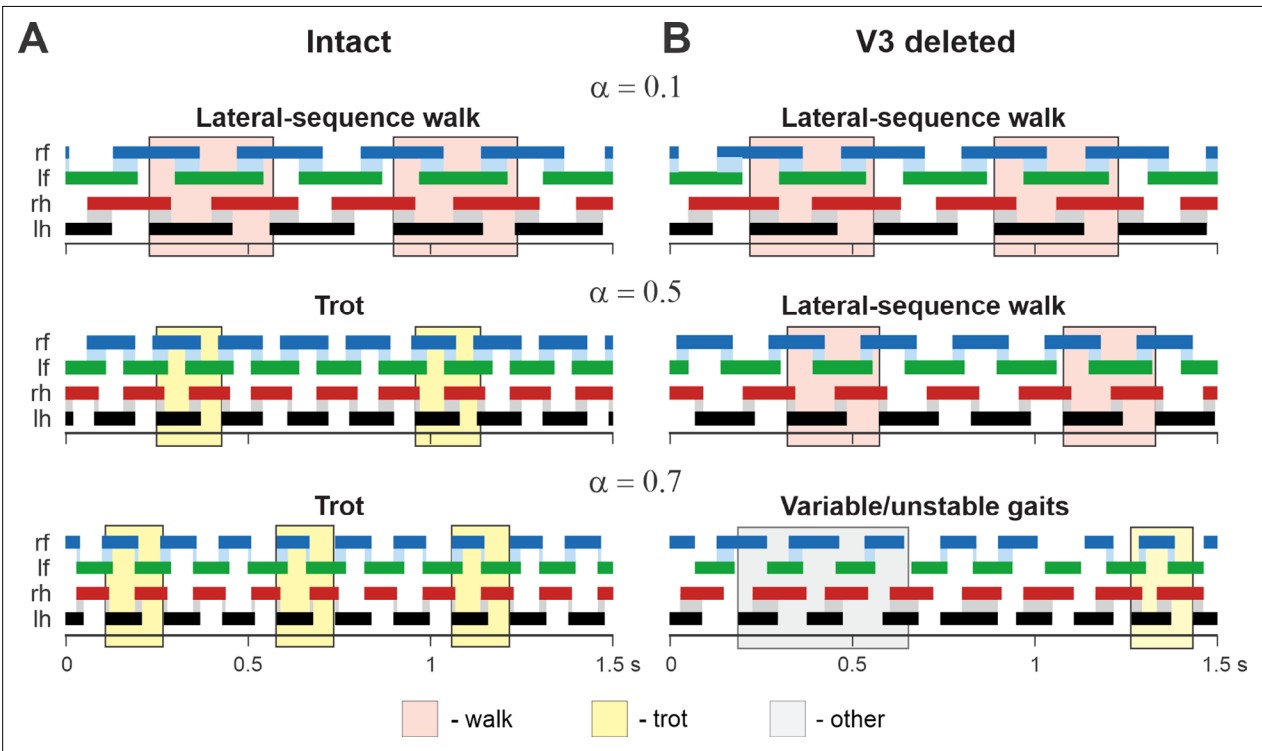

**Figure 12.** Examples of the step diagrams for the intact model (**A**) and after all V3 neurons were deleted (**B**). For each rhythm generator (RG), the lengths of the corresponding extensor phases are shown for 1.5 s of the simulation time for three values of α = {0.1; 0.5; 0.7}. At α = 0.1, both intact and V3-deleted models demonstrate stable lateral-sequence walking gait. At α = 0.5, the intact model exhibits a trot, and the V3-deleted model demonstrates a lateral-sequence walk. At α = 0.7, the intact model has a stable trot while the V3-deleted model demonstrates unstable behavior. The solid color bars show the extensor phases of all four RGs: rf, right fore; lf, left fore; rh, right hind; lh, left hind. The black bars show the reference RG (lh). Blue and gray shadows highlight periods of overlap between extensor phases of the homologue RGs. The shading color indicates the gait (pink, lateral-sequence walk; yellow, trot; light gray, other).

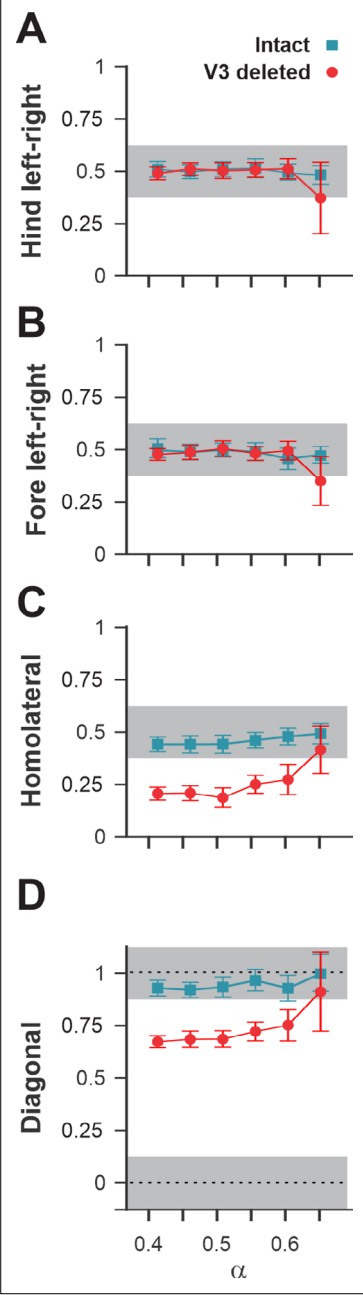

**Figure 13.** Interlimb coordination in the intact model and after removal of all V3 neurons. Interlimb coordination in the intact (blue) and V3-deleted (red) models for six values of the brainstem drive is shown for left–right hind (**A**), left–right fore (**B**), homolateral (**C**), and diagonal (**D**) normalized phase differences. For each value of $\alpha$, phase differences were averaged over 10–20 consecutive oscillation periods. In (**A**), (**C**), and (**D**), the extensor phase of the left hind rhythm generator (RG) is chosen as the reference. Note the homolateral and diagonal phase difference shift toward walking in V3-deleted model compared to trotting gate in the intact case.

contribution of the individual subpopulations of V3 neurons (local CINs and aLPNs) to the speed-dependent control of interlimb coordination and gait expression. The proposed ascending long propriospinal connections mediated by V3 aLPNs support diagonal synchronization necessary for trot, whereas the local V3 commissural connections support left–right synchronization necessary for gallop and bound. The proposed model is able to reproduce the experimentally observed speed-dependent gait expression of WT mice as well as changes in interlimb coordination following V3 silencing. The model strongly suggests the crucial and unique roles of V3 aLPNs in control of interlimb coordination during locomotion. Finally, the model provides a potentially testable prediction for the removal of only the V3 aLPNs subset, which broadens our understanding of the locomotor circuits and will guide our future experiments.

## Lumbar V3 neurons with ascending long propriospinal projections to the contralateral cervical region

The proper coordination between limb movements during locomotion, expressed as locomotor gaits, and adaptive gait changes is essential for animals to maintain stable movement within a wide range of locomotor speeds (*Hildebrand, 1976*; *Hildebrand, 1980*; *Hildebrand, 1989*). Each limb is primarily controlled by separate RG circuits located in a particular spinal cord compartment (*Forssberg et al., 1980*; *Frigon, 2017*). The central interlimb coordination is provided by multiple populations of spinal neurons that mediate mutual interactions between RG circuits controlling different limbs (*Stein, 1976*; *Schöner et al., 1990*; *Danner et al., 2016*; *Danner et al., 2017*; *Frigon, 2017*). These neuronal populations include local CINs, operating within the cervical and lumbar enlargements and coordinating movements of left and right forelimbs and left and right hindlimbs, respectively, as well as LPNs coordinating movements of forelimbs and hindlimbs.

This study focused on V3 neurons, which are excitatory neurons predominately projecting to the contralateral side of the spinal cord (*Zhang et al., 2008*). Until now, these neurons have been primarily considered within lumbar spinal circuits. In our previous experimental and modeling studies, we suggested that V3 neurons mediate mutual excitation between left and right RG circuits and promote the left–right synchronization needed for gallop and bound (*Rybak et al.,*

*2013*; *Rybak et al., 2015*; *Shevtsova et al., 2015*; *Danner et al., 2016*; *Danner et al., 2017*; *Danner et al., 2019*). However, since lumbar V3 neurons diverge into several subpopulations distinguished by their distribution, morphology, and electrophysiological properties (*Borowska et al., 2013*; *Borowska et al., 2015*; *Blacklaws et al., 2015*), it is likely that different V3 subpopulations perform different functions.

Here, we revealed a subpopulation of lumbar V3 neurons with long ascending projections to the contralateral cervical region. Although we cannot exclude the possibility that some of the CTB-labeled axons had supraspinal projections, our electrophysiological and anatomical studies have confirmed the existence of lumbar V3 neurons that directly innervate and activate cervical neuronal circuits (V3 aLPNs). Our data show that photoactivation of lumbar V3 neurons resulted in a motor response in the cervical cord. We suggest that this response was provided by projections of V3 aLPN neurons. It could also be that the light activated local (non-aLPN) lumbar V3 neurons that in turn activated non-V3 lumbar aLPNs projecting to the cervical cord. However, we also showed that the effect of lumbar V3 stimulation on the activity of cervical roots persisted after blockade of glutamate receptors in the lumbar cord. Taken together with our anatomical results, these experiments support the conclusion that the lumbar cord contains at least one subpopulation of aLPNs, whose activation leads to a motor response in the cervical cord. This, however, does not exclude the possible existence of additional ascending pathways, not involving V3 aLPNs, like the one described above, which have not been considered in our model.

While we have not identified the exact neuronal targets of V3 aLPNs in the cervical cord, we showed that a relatively large proportion of V3 ascending terminals is present in cervical lamina VII/VIII, the region where RG intraneuronal circuits are located. Furthermore, the recruitment of V3 aLPNs increased with an increase of locomotor speed, suggesting their engagement during locomotion and an increase of their involvement with an increase in locomotor speed. Our model reproduced this recruitment of V3 aLPNs. Based on our modeling results, we suggest that V3 aLPNs mediate excitation from each lumbar flexor half-center to the contralateral cervical flexor half-center, which promotes diagonal synchronization necessary for trot.

## The V3 neurons are necessary for high-speed locomotion, and their removal significantly limits the locomotor speed due to distortion of limb coordination

In our study, WT mice were able to locomote on the treadmill with high speeds up to 68–83 cm/s. Silencing V3 neurons led to a significant reduction of maximal locomotor speed. Our analysis of V3$^{OFF}$ mouse locomotion has shown a lack of diagonal synchronization at medium speeds and an increase of variability of left–right coordination with increasing locomotor speed. Similarly, removal of all V3 neurons from our model (simulating V3$^{OFF}$ mice) reduced the maximal frequency of stable locomotor oscillation. Model analysis revealed that this frequency reduction occurred due to the distortion of coordination between left–right homologous RGs and between diagonal lumbar and cervical RGs, both of which were mediated by V3 neurons. Yet, selective deletion of only V3 aLPNs in the model allowed for stable coordination of limb activities at high speeds, that is, during gallop and bound, whereas trot was completely lost, and the model transitioned from walk directly to gallop and bound. Together, our experimental and modeling results suggest that the significant reduction of locomotor speed in V3$^{OFF}$ mice results from the distortion of interlimb coordination, particularly the properly synchronized couplings, which are necessary for trot, gallop, and bound gaits that are normally used by WT mice at these speeds.

## The V3 neurons are involved in speed-dependent gait changes and are necessary for the expression of trot, gallop, and bound

A precise yet flexible control of interlimb coordination in a wide range of speeds allows animals to maintain dynamic stability in a continuously changing environment. This interlimb coordination, expressed as gaits, depends on, and changes with, locomotor speed and is controlled by the locomotor circuits in the spinal cord (*Kiehn, 2016*). Mice sequentially change gait from walk to trot and then to gallop and bound as locomotor speed increases (*Clarke and Still, 1999*; *Herbin et al., 2004*; *Herbin et al., 2007*; *Batka et al., 2014*; *Bellardita and Kiehn, 2015*; *Lemieux et al., 2016*). WT mice use walking gaits only during low-speed exploratory locomotion and generally prefer trot for

overground and treadmill locomotion (*Gruntman et al., 2007*; *Bellardita and Kiehn, 2015*; *Lemieux et al., 2016*; *Caggiano et al., 2018*). Gallop and bound are mainly used for escape behavior and are expressed only at high speeds (i.e., >75 cm/s) during both overground and treadmill locomotion (*Bellardita and Kiehn, 2015*; *Lemieux et al., 2016*).

Our WT mice showed similar preference of the trot gait at all locomotor speeds tested (15–40 cm/s). The maximal speed investigated is below the minimal speed at which mice gallop or bound and thus these gaits were not expected to occur. In contrast to the WT mice, the V3$^{OFF}$ mice preferred a lateral-sequence walk at almost all treadmill speeds, at which they could maintain stable limb coordination (up to 30 cm/s). At treadmill speeds close to their maximal speed (35–40 cm/s), few individual steps that could be classified as trot. At these higher speeds, left–right coordination became variable and even transient episodes with left–right synchronized steps were present (*Figure 5B2*). Similar gait changes in left–right coordination were observed after silencing dLPNs (*Ruder et al., 2016*) and aLPNs of unspecific genetic type (*Pocratsky et al., 2020*). Our previous computational model (*Danner et al., 2017*) attributed the speed-dependent disruption of left–right coordination after dLPN silencing (*Ruder et al., 2016*) to excitatory, diagonally projecting LPNs.

The specific function of V3 aLPNs would be best determined using selective activation or silencing of these neurons. Unfortunately, we do not currently have experimental tools to specifically and exclusively target the V3 aLPNs. Therefore, we utilized a network arrangement that included a population of V3 aLPNs and reproduced data in the WT and V3$^{OFF}$ conditions to determine the effects of selective removal of the V3 aLPN population. Our simulations showed that the V3 aLPN populations are crucial in mediating diagonal synchronization, which is essential for trot. The local V3 CINs mediated left–right synchronization necessary for gallop and bound. The present model retained the connectivity of several other neuron types (including V0$_V$, V0$_D$, V2a, and Shox2 neurons) from the previous model and reproduced speed-dependent gait expression (from walk to trot and then to gallop and bound) of WT mice. Elimination of all V3 populations in the model, performed to simulate locomotion of V3$^{OFF}$ mice, extended the frequency range of the lateral-sequence walk at which the model was still able to demonstrate stable locomotor activity, eliminated trot as well as high-speed synchronous gaits, and caused variable interlimb coordination at higher speeds. Moreover, like our experimental results, the conversion of trot to lateral-sequence walk at medium locomotor frequencies resulted from a shift of both the homolateral and the diagonal phase differences. These simulation results were fully consistent with our experimental finding described above.

In summary, our experimental and modeling results allow the suggestion that the V3 neurons critically contribute to speed-dependent gait expression and are necessary for the expression of trot, gallop, and bound.

## Modeling predictions and limitations of the model

While this study was mainly focused on V3 neurons, the importance of the proposed model would be low if the model could only reproduce data regarding the manipulation of V3 neurons described here. Our ultimate goal has always been to develop a generalized model that would be consistent with a large body of data obtained in multiple laboratories under different experimental conditions. Major parts of the network architecture used in the present model were based on our previous models (*Rybak et al., 2015*; *Shevtsova et al., 2015*; *Danner et al., 2017*; *Danner et al., 2019*), which are able to reproduce experimental data from multiple studies using selective removal, silencing, inhibition, or activation of different types of genetically identified spinal interneurons. Using these previous models as a basis, we were very careful to make sure that the updated model presented here is still able to reproduce the previous experimental data. Therefore, despite the relative complexity of the model, the number of different experimental phenomena that the final model can reproduce provides additional validation of the proposed model.

Since much of the architectural/structural data, including known types of neurons and their connectivity are unknown, the computational model presented here relies on multiple assumptions, which were specifically made to reproduce a series of known behaviors. Thus, these suggested elements (neuron types) and connectivity represent the modeling predictions, which should then be tested in the future to validate the model.

The current model includes two types of V3 neurons: local V3 CINs that operate within lumbar or within cervical compartments providing mutual excitation between the homonymous left and right

RGs, and V3 aLPNs mediating diagonal excitation from each lumbar flexor half-center to the contralateral cervical flexor half-center. The local V3 CINs promote left–right synchronization necessary for gallop and bound, and the V3 aLPNs promote diagonal synchronization necessary for trot. Therefore, we suggest that the spinal cord contains two types of V3 subpopulations that have distinct functions. Specifically, the V3 aLPNs are necessary for the expression of trot at intermediate locomotor speeds, whereas the local V3 CINs are critical for gallop and bound at high locomotor speeds. This suggestion could be tested by silencing only V3 aLPNs: simulations predicted that this would result in a loss of trot only; gallop and bound would remain stable at higher locomotor speeds.

It is also interesting to note that based on our present model stable trot requires two necessary conditions: (i) left–right alternation in the lumbar and cervical compartments, mediated by V0 (V0$_D$ and V0$_V$, *Talpalar et al., 2013*; *Bellardita and Kiehn, 2015*; *Shevtsova et al., 2015*; *Rybak et al., 2015*) CINs, and (ii) diagonal synchronization mediated by V3 aLPNs (aV3 in the model). This invalidates our previous modeling prediction that removal of V3 neurons should extend trot to locomotion at high speeds (by replacing gallop and bound; *Danner et al., 2016*) and supports the current conclusion that the removal of all V3 distorts stable trot, limit locomotor frequency and allows only walk at low to medium frequencie.

The other interesting feature of the model is the presence of the diagonal dLPN pathways mediated by the inhibitory V0$_D$ neurons (*Figures 8 and 9B*). The V0$_D$ dLPNs in the model perform the opposite function of the V3 aLPNs: they promote diagonal alternation between the cervical and lumbar RGs and hence suppress trot and promote walk at low locomotor frequencies (and speeds). The removal of these V0$_D$ neurons would make the lateral-sequence walk at low speeds unstable, which is partly supported by the experimental data in vitro (*Talpalar et al., 2013*; *Kiehn, 2016*). To reproduce this behavior, our model includes synaptic inhibition of V0$_D$ dLPNs by V3 aLPNs via intermediate inhibitory populations (In2; *Figures 8 and 9C*). These inhibitory connections provide a stable transition from walk to trot during locomotion at slow to medium speeds. These proposed connections can be considered as an additional modeling prediction awaiting future experimental testing.

Considering the complexity of the proposed model and current insufficiency of experimental data, the proposed model has multiple limitations. One such limitation is that the present model focuses exclusively on central interactions within the spinal cord, without considering biomechanics and the role of sensory feedback from the limbs, which are involved in limb coordination and gait expression in vivo. Therefore, some of the features in the model that are implemented through central neural interactions may in reality result from interactions between limb or body biomechanics and neuronal circuits. Indeed, V3 neurons are likely involved in the integration of afferent feedback. Most V3 aLPNs are located across the deep dorsal horn and intermediate region of the rostral lumbar spinal cord. Our previous study showed that V3 neurons in these regions are highly active during overground activity, but silent during swimming (*Borowska et al., 2013*). Neurons in this region are also known to receive intensive innervation from group Ib, group II, and skin sensory afferents (*Edgley and Jankowska, 1987*; *Bannatyne et al., 2009*; *Jankowska and Edgley, 2010*). Therefore, the lumbar V3 aLPNs may be involved in mediating and integrating signals from both the lumbar locomotor RGs and sensory information from hindlimb muscles and joints to regulate the coordination of forelimb and hindlimb movements during locomotion. Future studies may consider afferent feedback interactions with V3 neurons and other parts of the spinal locomotor circuitry by integrating the neural network models with a model of the musculoskeletal system (*Nishikawa et al., 2007*; *Markin et al., 2015*; *Prilutsky and Edwards, 2016*; *Ausborn et al., 2021*) to simulate interactions between the neural circuits, biomechanical constraints, and the environment.

The other important limitation is that our model did not consider spinal circuits operating below the RG and limb-coordinating circuits. The model does not include motoneurons, and we assume that the output motor activity (recorded from the lumbar and cervical roots) simply reproduces the output activity in rhythm generating circuits. Therefore, different pattern formation circuits, circuits involved in the processing of sensory feedback, and reflex circuits, including those mediating by Ia and Ib interneurons, Renshaw cells, and motoneurons (see *Rossignol et al., 2006*; *Rybak et al., 2006*; *Alvarez and Fyffe, 2007*; *McCrea and Rybak, 2007*; *McCrea and Rybak, 2008*; *Pierrot-Deseilligny and Burke, 2012*; *Zhong et al., 2012*), were not included in the model. These circuits play an important role in intralimb coordination and can also affect interlimb coordination but have not been considered

in our present modeling study. In the future modeling studies, we will focus on sequential reduction of the above limitations.

Nevertheless, here, we took advantage of available experimental and computational tools to demonstrate that two subgroups of lumbar V3 neurons, which possess local projections or long ascending propriospinal commissural projections, are essential components in the locomotor circuits that promote speed-dependent interlimb synchronization to provide stable locomotion at different speed and gaits.

# Materials and methods
## Experimental materials and methods
### Animals
The generation and genotyping of $Sim1^{Cre/+}$ mice were described previously by *Zhang et al., 2008*. Conditional knock-out of Vglut2 in Sim1-expressing V3 INs, $Sim1^{cre/+}$; $Slc17a6^{flox/flox}$ (V3$^{OFF}$ mice) was described previously by *Chopek et al., 2018*. Ai32 mice ($Gt(ROSA)26^{floxstopH134R/EYFP/+}$, Jackson Laboratory, Stock No. 012569) were crossed with $Sim1^{Cre/+}$ to generate $Sim1^{Cre/+}$; Ai32 mice. TdTomato Ai9 mice ($Rosa26^{floxstopTdTom}$, Jackson Laboratory, Stock No. 007909) were crossed with $Sim1^{Cre/+}$ to generate $Sim1^{Cre/+}$; $Rosa26^{floxstopTdTom}$ (Sim1TdTom) mice (*Blacklaws et al., 2015*). All procedures were performed in accordance with the Canadian Council on Animal Care and approved by the University Committee on Laboratory Animals at Dalhousie University.

### Treadmill locomotion
Treadmill locomotion tests were performed using 11 (six males, five females) V3$^{OFF}$ ($Sim1^{cre/+}$; $Slc17a6^{flox/flox}$) mice and 7 (two males, five females, $Sim1^{+/+}$; $Slc17a6^{flox/flox}$ or $Sim1^{+/+}$; $Slc17a6^{+/flox}$) control littermates (WT) at postnatal days 40–48 (P40–48). No training was performed before any locomotion tests. During the locomotion tests, the mice were subjected to a treadmill, Exer Gait XL (Columbus Instruments), at speeds from 15 cm/s to 40 cm/s in 5 cm/s increments. To avoid their fatigue, animals would not perform more than two trials for each speed during each experiment, and each trail was <20 s. There was a >1 min rest period between consecutive trials.

A mirror was placed underneath the transparent treadmill belt with 45° angle to project the image of the paws to a high-speed camera. The paw movements were captured at 100 frames/s. Episodes with at least 10 consecutive steps were included for analysis. For the maximum speed test, if the mouse failed to hold the speed for more than 3 s after five tries, this speed was defined as the maximum speed for this mouse.

### Gait analysis
The movement of four paws in the videos was manually tracked using Vicon Motus software. The foot contact was defined as the time of the first frame in three consecutive frames in which the size of the image of the paw on the ground did not change. The foot lift was defined as the time of the frame when the front part of the paw disappeared from the image. The time sequences of the paw movement were exported and processed using a custom-written script in Spike2 (version 7.09a, Cambridge Electronic Design). The stride duration was defined as the duration between two consecutive foot contacts of the same paw. A stance phase started at the foot contact and ended at the foot-lift. A swing phase was the period from the offset of stance to the onset of the next stance. The normalized phase differences between the limbs were calculated as the time difference between stance onset of the tested limb and the reference limb (left hindlimb; lh) divided by the step-cycle duration of the reference limb (*Figure 5A*). All normalized phase differences ranged from 0 to 1. The value of 0 or 1 indicated a perfect in-phase coupling (synchronization), while 0.5 indicated a perfect anti-phase coupling (alternation). We classified trot, lateral-sequence walk (L-walk), bound, and out-of-phase walk (OPW) gaits as described in *Lemieux et al., 2016*. All other gaits defined by *Lemieux et al., 2016* were categorized as 'others.'

To describe the role of different gaits during treadmill locomotion, we calculated their occurrence, persistence, and attractiveness in all steps (*Lemieux et al., 2016*). The occurrence was the percentage of a given gait within the total steps of a trial. The persistence of a certain gait was the percentage of

two consecutive steps using the same gait out of all the consecutive steps. The attractiveness was the possibility of other gaits transferring to the focused gait.

## CTB and AAV-GFP injections

To identify and characterize V3 aLPNs, we did two microinjections: (i) CTB conjugated with Alexa Fluor 488 (0.5 mg/1 ml PBS, Thermo Fisher) was injected at C5 to C8 segments. Briefly, mice were anesthetized under isoflurane throughout the surgery. A 1 cm sagittal incision was made centered over the sixth cervical spinous process for cervical injection or over the spine around the last rib for lumbar injection. A laminectomy was performed, and the bone covering the lower fifth, sixth, seventh, and upper eighth cervical segments (caudal C5 to rostral C8) was removed. The glass micropipette filled with the CTB at the tip was lowered into the spinal cord. The manipulator adjusted the position of the micropipette. Using a Nanoject II (Drummond), 500–700 nl of CTB was injected at multiple locations on one side of the spinal cord. The pipette was kept in place for additional 5 min for diffusion. After injection, the muscles were sewn back together in layers with absorbable sutures, and the skin was sewn closed with polypropylene surgical sutures. The spinal cords were harvested 7 days after CTB injections. (ii) AAV2/9-hSyn-eGFP (Molecular Tools Platform, Canada) was injected at L1–L3 segments of P30–35 Sim1TdTom mice. The procedures are the same as described above, but for the lumbar injection, the laminectomy was performed on the 11th and 12th thoracic vertebrae to uncover the L1–L3 spinal cord segments, and AAVs were injected to both sides of the spinal cord. The spinal cords were harvested 3 weeks after the AAV injections.

## Immunohistochemical procedures and confocal imaging

To detect the c-Fos expression, after 7 days for the CTB transport, the animals were subjected to walk (15 cm/s) or run (40 cm/s) on a treadmill for three times 15 min with a 5 min interval between trials. The animals in the control group were left in the home cage through the experiments. After the task, mice were put back into the home cage for 60 min prior to perfusion and tissue extraction.

To harvest spinal cords for immunohistochemistry, the animals were perfused with 4% paraformaldehyde (PFA) in phosphate-buffered saline (PBS). Spinal cords were extracted and postfixed in 4% PFA at 4°C for 3–4 hr and then cryoprotected in 30% sucrose for 2–3 nights at 4°C before embedding in OCT and cryostat sectioning. 30-μM-thick sections were collected for immunolabeling. Sections were washed in PBS with 0.1% Triton X-100 (PBS-T) three times for 15–20 min each rinse. Sections were the incubated in 10% goat serum PBS with primary antibodies at 4°C for one or two (c-Fos) nights. Sections were then washed in 3× PBS (15–20 min each wash) and incubated with secondary antibodies for 1 hr at 4°C. Sections were then washed in PBS three times and cover-slipped with an anti-fade mounting medium (Dako). Primary antibodies used in this study: rabbit anti-cFos antibody (1:2000; Sysy); goat anti-dsRed antibody (1:1000; Acris Antibodies, Cat# AB8181-200), guinea pig anti-Vglut2 (1:1000; MilliporeSigma, Cat# AB2251-I), and chicken anti-GFP (1:500; Aves Labs, Cat# GFP-1010). Secondary antibodies: goat anti-rabbit conjugated to Alexa Fluor 488 (1:500; The Jackson Laboratory), donkey anti-goat conjugated to Alexa Fluor 594 (1:500; The Jackson Laboratory), donkey anti-guinea pig conjugated to DyLight 405 (1:500; The Jackson Laboratory), and donkey anti-chicken conjugated to Alexa Fluor 488 (1:500, The Jackson Laboratory).

Confocal images were obtained using a Zeiss LSM 880 laser scanning confocal microscope with a Zeiss Plan-Apochromat ×10, ×20, or ×63 objective lens utilizing the tiling function of Zeiss ZEN Pro imaging software. The images were processed using Imaris 8.1.2 software. Cells with triple- or double-positive fluorescent signals were manually detected and mapped. In addition to using the central canal and Rexed's laminae as landmarks, we also set grids on the image of transverse sections of the spinal cord and the schematic section to map the double-labeled cells more accurately.

To sample synaptic terminals in the cervical region, ×63 objective lens and the tiling function of Zeiss ZEN Pro imaging software were used, specifically 2 vertical × 3 horizontal (~370 μm × 480 μm) in lamina IX and 2 × 2 (~370 μm × 370 μm) in lamina VII/VIII. Images were analyzed using Imaris 8.1.2 software. The 'spots' function in Imaris was used to identify individual tdTom[+] contacts, and these contacts were manually and individually analyzed to identify dual-labeled Vglut2[+]/AAV[+] and triple-labeled tdTom[+]/Vglut2[+]/AAV[+] contacts.

## Electrophysiology

All experiments were performed using spinal cords from Sim1Ai32 mice at P2–P3. The mice were anesthetized, and the spinal cords caudal to C1 were dissected out in Ringer's solution at room temperature (111 mM NaCl, 3.08 mM KCl, 11 mM glucose, 25 mM NaHCO$_3$, 1.25 mM MgSO$_4$, 2.52 mM CaCl$_2$, and 1.18 mM KH$_2$PO$_4$, pH 7.4). The spinal cord was then transferred to the recording chamber to recover at room temperature for at least 1 hr before recording in Ringer's solution. The recording chamber was partitioned by a narrow petroleum jelly (Vaseline) bridge into two parts with independent perfusion systems. The T6–T8 spinal segments were in the petroleum jelly and the whole lumbar (L) and cervical (C) region were exposed in the bath. A dye was added into one of the compartments to check the water tightness in the end of experiment. ENG recordings of the L5 and C5/C8 ventral roots were conducted using differential AC amplifier (A-M system, model 1700) with the band-pass filter between 300 Hz and 1 kHz. Analog signals were transferred and recorded through the Digidata 1400 A board (Molecular Devices) under the control of pCLAMP10.3 (Molecular Devices).

To activate ChR2 in V3 INs, 488 nm fluorescent light was delivered by Colibri.2 illumination system (Zeiss) through 10 × 1.0 numerical aperture (NA) objectives mounted on an upright microscope (Examiner, Zeiss) onto the ventral surface of L1–L3 or C5–C7 segments of the isolated spinal cord. Continuous light stimuli with duration of 200 ms were used. The stimulation was applied 12 times. Then, 2 mM KA (Sigma-Aldrich) was added into the compartment for the lumbar spinal cord. The same optical stimulation to the same area was applied during the drug application and after the washout.

## Statistical analysis

Statistical analysis was performed in Prism 7 (GraphPad Software, Inc) and MATLAB (version R2018a, The MathWorks, Inc). Brown–Forsythe ANOVA test was used to compare the ENG activity change in the same root among different conditions. Kolmogorov–Smirnov test was used to compare the difference of maximum speed between WT and V3$^{OFF}$ mice. Welch's $t$-test was used to compare the difference of c-Fos expression in Sim1 cells and Sim1/CTB cells among different conditions. Watson–Williams test of Homogeneity of Means was used to compare phase differences between WT and V3$^{OFF}$ mice across all speeds and at specific speeds. The Equal Kappa test was used to compare the concentration (a measure of variability) of the phase differences between WT and V3$^{OFF}$ mice across all speeds and at specific speeds. p-Values of all post-hoc tests (comparisons at specific treadmill speeds) were adjusted using Bonferroni's method.

## Modeling methods

### Single neuron

All neurons were simulated in the Hodgkin–Huxley style as single-compartment models. The membrane potential, $V$, in neurons of the flexor (F) and extensor (E) half-centers is described as

$$C \times \frac{d}{dt}V = -I_{Na} - I_{NaP} - I_K - I_L - I_{SynE} - I_{SynI}, \tag{1}$$

where $C$ is the membrane capacitance; $t$ is time; $I_{Na}$ is the fast Na$^+$ current; $I_{NaP}$ is the persistent (slowly inactivating) Na$^+$ current; $I_K$ is the delayed-rectifier K$^+$ current; $I_L$ is the leakage current; and $I_{SynE}$ and $I_{SynI}$ are synaptic excitatory and inhibitory currents.

In all other populations, the neuronal membrane potential is described as follows:

$$C \times \frac{d}{dt}V = -I_{Na} - I_K - I_L - I_{SynE} - I_{SynI}. \tag{2}$$

The ionic currents in *Equations (1) and (2)* are described as follows:

$$
\begin{aligned}
I_{Na} &= \bar{g}_{Na} \times m_{Na}^3 \times h_{Na} \times (V - E_{Na}); \\
I_{NaP} &= \bar{g}_{NaP} \times m_{NaP} \times h_{NaP} \times (V - E_{Na}); \\
I_K &= \bar{g}_K \times m_K^4 \times (V - E_K); \\
I_L &= g_L \times (V - E_L);
\end{aligned}
\tag{3}
$$

where $\bar{g}_{Na}$, $\bar{g}_{NaP}$ (present only in RG neurons), $\bar{g}_K$ and $g_L$ are maximal conductances of the corresponding currents; $E_{Na}$, $E_K$, and $E_L$ are the reversal potentials for Na$^+$, K$^+$, and leakage currents,

respectively; variables $m$ and $h$ are the activation and inactivation variables of the corresponding ionic channels (indicated by the indices). The maximal conductances for ionic currents and the mean leak reversal potentials, $E_{L0}$, were defined as follows: in F and E half-centers, $\bar{g}_{Na}$ = 25 mS/cm², $\bar{g}_{NaP}$ = 0.8 mS/cm², $\bar{g}_K$ = 2 mS/cm², and $g_L$ = 0.18 mS/cm², $E_{L0}$ = −66.4 (±0.664) mV; in all other populations, $\bar{g}_{Na}$ = 10 mS/cm², $\bar{g}_K$ = 5 mS/cm², and $g_L$ = 0.1 mS/cm², $E_{L0}$ = −68( ± 2) mV.

Activation $m$ and inactivation $h$ of voltage-dependent ionic channels (i.e., Na, NaP, and K) in **Equation (3)** are described by the differential equations:

**Table 1.** Brainstem drive parameters.

| Target population | $k_i$ | $d_{0i}$ |
|---|---|---|
| *Excitatory drives* | | |
| RG-E | 0.0 | 0.5 |
| RG-F (f) | 0.15 | 0 |
| RG-F (h) | 0.3 | 0 |
| V3-F | 1 | 0 |
| aV3 | 1 | 0 |
| *Inhibitory drives* | | |
| V0$_V$ | -8 | 0 |
| V0$_D$ | −10 | 0 |
| Diagonal V0$_V$ | -3 | 0 |
| Diagonal V0$_D$ | -3 | 0 |

f, forelimb; h, hindlimb; RG, rhythm generator.

$$\tau_{mi}(V) \times \frac{d}{dt}m_i = m_{\infty i}(V) - m_i;$$
$$\tau_{hi}(V) \times \frac{d}{dt}h_i = h_{\infty i}(V) - h_i, \tag{4}$$

where $m_{\infty i}(V)$ and $h_{\infty i}(V)$ define the voltage-dependent steady-state activation and inactivation of the channel $i$, respectively, and $\tau_{mi}(V)$ and $\tau_{hi}(V)$ define the corresponding time constants. Activation of the sodium channels was considered instantaneous. The expressions for channel kinetics in **Equation (4)** are described as follows:

$$
\begin{aligned}
m_{\infty Na}(V) &= (1 + \exp(-(V + 34)/7.8))^{-1}; \tau_{mNa} = 0;\\
h_{\infty Na}(V) &= (1 + \exp((V + 55)/7))^{-1};\\
\tau_{hNa}(V) &= 20/(\exp((V + 50)/15) + \exp(-(V + 50))/16);\\
m_{\infty NaP}(V) &= (1 + \exp(-(V + 47.1)/3.1))^{-1}; \tau_{mNaP} = 0\\
h_{\infty NaP}(V) &= (1 + \exp((V + 50)/6.8))^{-1}; \tau_{hNaP}(V) = 180/\cosh((V + 50)/13.6);\\
m_{\infty K}(V) &= (1 + \exp(-(V + 28)/4))^{-1}; \tau_{mK}(V) = 3.5/\cosh((V + 40)/40); h_k = 1.
\end{aligned}
\tag{5}
$$

The synaptic excitatory ($I_{SynE}$ with conductance $g_{SynE}$ and reversal potential $E_{SynE}$) and inhibitory ($I_{SynI}$ with conductance $g_{SynI}$ and reversal potential $E_{SynI}$) currents are described as follows:

$$
\begin{aligned}
I_{SynE} &= g_{SynE} \times (V - E_{SynE});\\
I_{SynI} &= g_{SynI} \times (V - E_{SynI})
\end{aligned}
\tag{6}
$$

where $g_{SynE}$ and $g_{SynI}$ are equal to zero at rest and are activated by the excitatory or inhibitory inputs, respectively:

$$
\begin{aligned}
g_{SynEi}(t) &= \bar{g}_E \times \sum_j S\{w_{ji}\} \times \sum_{t_{kj}<t} \exp\left(-(t - t_{kj})/\tau_{SynE}\right) + \bar{g}_{Ed} \cdot S\{w_{di}\} \cdot d_i;\\
g_{SynIi}(t) &= \bar{g}_I \times \sum_j S\{-w_{ji}\} \times \sum_{t_{kj}<t} \exp\left(-(t - t_{kj})/\tau_{SynI}\right) + \bar{g}_{Id} \cdot S\{-w_{di}\} \cdot d_i,
\end{aligned}
\tag{7}
$$

where $S\{x\} = x$, if $x \geq 0$, and 0 if $x < 0$. In **Equation 7**, the excitatory and inhibitory synaptic conductance have two terms: one describing the effects of excitatory or inhibitory inputs from other neurons in the network and the other describing effects of inputs from the external brainstem excitatory or inhibitory drives (see also **Rybak et al., 2006**). Each spike arriving to neuron $i$ in a target population from neuron $j$ in a source population at time $t_{kj}$ increases the excitatory synaptic conductance by $\bar{g}_E \times w_{ji}$ if the synaptic weight $w_{ji} > 0$ or increases the inhibitory synaptic conductance by $-\bar{g}_E \times w_{ji}$ if the synaptic weight $w_{ji} < 0$. $\bar{g}_E$ and $\bar{g}_I$ define an increase in the excitatory or inhibitory synaptic conductance, respectively, produced by one arriving spike at $|w_{ji}| = 1$. $\tau_{SynE}$ and $\tau_{SynI}$ are the decay time constants for $g_{SynE}$ and $g_{SynI}$ , respectively. In the second terms of **Equation 7**, $\bar{g}_{Ed}$ and $\bar{g}_{Id}$ are the parameters defining the increase in the excitatory or inhibitory synaptic conductance, respectively, produced by external input drive $d_i = 1$ with a synaptic weight of $|w_{di}| = 1$.

Excitatory and inhibitory drives to population   were modeled as a linear function of the free parameter $\alpha$:

$$d_i\left(\alpha\right) = k_i \cdot \alpha + d_{0i} + d_{\text{noise}}, \tag{8}$$

where $k_i$ is the slope, $d_{0i}$ is the intercept, and $\alpha$ is the strength of brainstem drive. The values of $k_i$ and $d_0$ for all populations receiving the brainstem drive are indicated in **Table 1**. For the calculation of the bifurcation diagrams (**Figure 11**), Gaussian noise with a mean of 0 and a standard deviation of 10% of the drive amplitude was added to the drive to characterize the model stability and facilitate transition between gaits in the areas of bi- and multistability.

The following general neuronal parameters were used: $C$ = 1 µF·cm$^{-2}$; $E_{\text{Na}}$ = 55 mV; $E_{\text{K}}$ = − 80 mV; $E_{\text{SynE}}$ = −10 mV; $E_{\text{SynI}}$ = −70 mV; $\bar{g}_{\text{E}} = \bar{g}_{\text{I}} = \bar{g}_{\text{Ed}} = \bar{g}_{\text{Id}}$ = 0.05 mS/cm$^2$; $\tau_{\text{SynE}} = \tau_{\text{SynI}}$ = 5 ms.

## Neuron populations

The F and E half-centers in the model had 200 neurons, all other populations incorporated 100 neurons. Random synaptic connections between the neurons of interacting populations were assigned prior to each simulation based on probability of connection, $p$, so that, if a population $A$ was assigned to receive an excitatory (or inhibitory) input from a population $B$, then each neuron in population $A$ would get the corresponding synaptic input from each neuron in population $B$ with the probability $p\{A, B\}$. If $p\{A, B\} < 1$, a random number generator was used to define the existence of each synaptic connection; otherwise, (if $p\{A, B\} = 1$) each neuron in population $A$ received synaptic input from each neuron of population $B$. Values of synaptic weights ($w_{ji}$) were set using random generator and were based on average values of these weights $\bar{w}$ and variances, which were defined as 5% of $\bar{w}$ for excitatory

**Table 2.** Connection weights.

| Source | Target ($w_{ij}$) |
|---|---|
| Within cervical and lumbar circuits | |
| i-F | i-F (0.0125, $P$ = 0.1); i-InF (0.3, $P$ = 0.1), i-V0$_D$ (0.3, $P$ = 0.1), i-V2a (0.3, $P$ = 0.1), i-V3-F (0.3, $P$ = 0.1) |
| i-E | i-E (0.0125, $P$ = 0.1); i-InE (0.2, $P$ = 0.1), i-V3-E (0.2, $P$ = 0.05), i-Sh2 (0.1, $P$ = 0.1) |
| i-InF | i-E (–0.6, $P$ = 0.1) |
| i-InE | i-F (–0.07, $P$ = 0.1) |
| i-V2a | i-V0$_V$ (0.4, $P$ = 0.1) |
| i- V0$_V$ | c-Ini1 (0.4, $P$ = 0.1) |
| i-Ini1 | i-F (–2, $P$ = 0.1) |
| i-V0$_D$ | c-F (–1, $P$ = 0.1) |
| i-V3-E | c-E (0.01, $P$ = 0.05), c-InE (0.2, $P$ = 0.05) |
| i-V3-F | c-F (0.03, $P$ = 0.1), i-Ini2 (0.05, $P$ = 0.1) |
| Within cervical circuits | |
| i-F | i-LPNi (0.4, $P$ = 0.1), diagonal i-V0$_D$ (0.25, $P$ = 0.1), diagonal i-V0$_V$ (0.25, $P$ = 0.1) |
| Within lumbar circuits | |
| i-F | i-aV3 (0.1, $P$ = 0.1) |
| Between cervical and lumbar circuits | |
| i-Sh2 (f) | ih-F (0.02, $P$ = 0.1) |
| i-Sh2 (h) | if-F (0.05, $P$ = 0.1) |
| i-LPNi (f) | ih-F (–0.2, $P$ = 0.1) |
| diagonal i-V0$_D$ (f) | ch-F (–1.2, $P$ = 0.1) |
| diagonal if-V0$_V$ (f) | ch-F (0.1, $P$ = 0.1) |
| i-aV3 | cf-F (0.12, $P$ = 0.1), c-Ini2 (0.5, $P$ = 0.1) |

i, ipsilateral; c, contralateral; f, forelimb; h, hindlimb.

connections ($\bar{w} > 0$) and 10% of $w$ for inhibitory connections ($\bar{w} < 0$). The average weights and probabilities of connections are specified in *Table 2*.

Heterogeneity of neurons within each population was provided by random distributions of the mean leakage reversal potentials $\bar{E}_{L0}$ and initial conditions for the values of membrane potential and channel kinetics variables. The values of $E_{L0}$ and all initial conditions were assigned prior to simulations from their defined average values and variances using a random number generator. A settling period of 1–5 s was allowed in each simulation to stabilize the model variables.

## Computer simulations

All simulations were performed using the custom neural simulation package NSM 2.5.6. The simulation package and model configuration file to create the simulations presented in this article are available at https://github.com/RybakLab/nsm, (copy archived at swh:1:rev:b66ab0ab77d02f1e-7fe639a4fa78231d4df58968; *Markin et al., 2022*) and https://github.com/RybakLab/nsm/tree/master/models/Zhang-Shevtsova-2022. The simulation package was previously used for the development of several spinal cord models (*Rybak et al., 2006*, *Rybak et al., 2013*; *McCrea and Rybak, 2007*; *McCrea and Rybak, 2008*; *Zhong et al., 2012*; *Shevtsova et al., 2015*; *Shevtsova and Rybak, 2016*). Differential equations were solved using the exponential Euler integration method with a step size of 0.1 ms. Simulation results were saved as ASCII files containing time moments of spikes for all RG populations.

## Data analysis in computer simulations

The simulation results were processed using custom MATLAB scripts (The MathWorks, Inc, MATLAB 2020b). To assess the model behavior, the averaged integrated population activities (average number of spikes per neuron per second) were used to determine the onsets and offsets of bursts and calculate the burst durations for particular population. The timing of onsets of the populational bursts was determined at a threshold level equal to 10% (for RG populations) or 30% (for V3 populations) of the average difference between maximal and minimal burst amplitude for a particular population in the current simulation. The locomotor period was defined as the duration between two consecutive onsets of the extensor bursts of the reference (left hind) RG. Duration of individual simulations depended on the value of parameter $\alpha$, and to robustly estimate average values of burst duration and oscillation, for each value of $\alpha$, the first 10–20 transitional cycles were omitted to allow stabilization of model variables.

Normalized phase differences for each cycle were calculated as the durations between the onsets of the population burst and the onsets of the extension phase of the reference RG divided by the period. Gait classification was performed as for the experimental data using the definitions of *Lemieux et al., 2016*. To evaluate possible gaits with increasing brainstem drive, parameter $\alpha$ was linearly increased, and for each value of $\alpha$, the average frequency and left–right, homolateral, and diagonal phase differences were calculated. The frequency and phase differences were then plotted against the parameter $\alpha$ (*Figures 11 and 13*) or shown as circular plot diagrams (*Figure 11—figure supplement 1*).

## Acknowledgements

The authors thank Dr. Ole Kiehn for useful comments and suggestions and Dr. Joanna Borowska-Fielding, Mr. Dallas Bennett, Mr. Ramez Michail, and Mr. Igor Tatarnikov for technical support. This work was supported by the grants of the National Institutes of Health: R01 NS104194 (KJD), R01 NS110550 (IAR), R01 NS112304 (SMD), R01 NS115900 (SMD), and R21 NS118226 (KJD); the National Science Foundation: grant # 2113069 (IAR); the Canadian Institutes of Health Research: MOP110950 (YZ) and PJT-173547 (YZ); and the Natural Sciences and Engineering Research Council of Canada: RGPIN 04880 (YZ).

## Additional information

### Funding

| Funder | Grant reference number | Author |
| --- | --- | --- |
| National Institutes of Health | R01NS104194 | Kimberly J Dougherty |
| National Institutes of Health | R21NS118226 | Kimberly J Dougherty |
| National Institutes of Health | R01NS112304 | Simon M Danner |
| National Institutes of Health | R01NS115900 | Simon M Danner |
| National Institutes of Health | R01NS110550 | Ilya A Rybak |
| National Science Foundation | 2113069 | Ilya A Rybak |
| Canadian Institutes of Health Research | MOP110950 | Ying Zhang |
| Canadian Institutes of Health Research | PJT-173547 | Ying Zhang |
| Natural Sciences and Engineering Research Council of Canada | RGPIN 04880 | Ying Zhang |

The funders had no role in study design, data collection and interpretation, or the decision to submit the work for publication.

### Author contributions

Han Zhang, Data curation, Formal analysis, Investigation, Methodology, Resources, Validation, Visualization, Writing – original draft, Writing – review and editing; Natalia A Shevtsova, Conceptualization, Formal analysis, Investigation, Methodology, Software, Validation, Visualization, Writing – original draft, Writing – review and editing; Dylan Deska-Gauthier, Data curation, Investigation, Resources, Writing – review and editing; Colin Mackay, Investigation, Resources; Kimberly J Dougherty, Methodology, Validation, Writing – original draft, Writing – review and editing; Simon M Danner, Conceptualization, Formal analysis, Funding acquisition, Investigation, Methodology, Software, Validation, Visualization, Writing – original draft, Writing – review and editing; Ying Zhang, Conceptualization, Data curation, Funding acquisition, Project administration, Resources, Supervision, Writing – original draft, Writing – review and editing; Ilya A Rybak, Conceptualization, Funding acquisition, Methodology, Project administration, Software, Supervision, Visualization, Writing – original draft, Writing – review and editing

### Author ORCIDs

Han Zhang http://orcid.org/0000-0001-5494-3504
Natalia A Shevtsova http://orcid.org/0000-0002-1971-9707
Kimberly J Dougherty http://orcid.org/0000-0002-0807-574X
Simon M Danner http://orcid.org/0000-0002-4642-7064
Ying Zhang http://orcid.org/0000-0003-4363-5666
Ilya A Rybak http://orcid.org/0000-0003-3461-349X

### Ethics

All procedures were performed in accordance with the Canadian Council on Animal Care and approved by the University Committee on Laboratory Animals at Dalhousie University.

### Decision letter and Author response

Decision letter https://doi.org/10.7554/eLife.73424.sa1
Author response https://doi.org/10.7554/eLife.73424.sa2

## Additional files

### Supplementary files
• Supplementary file 1. Statistical results.

• Transparent reporting form

### Data availability
All data generated and analysed during this study are included in the manuscript and supporting file; source code for computational modeling is available from https://github.com/RybakLab/nsm, (copy archived at swh:1:rev:b66ab0ab77d02f1e7fe639a4fa78231d4df58968), and the model file is uploaded at https://github.com/RybakLab/nsm/tree/master/models/Zhang-Shevtsova-2022.

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
