## [Editor Report]

This article will interest neuroscientists who study how spinal circuits control locomotion. While the role of spinal interneurons in control of left–right and flexor–extensor alternations has been studied extensively, their role in hind–forelimb coordination has not been sufficiently studied. Zhang et al. study interlimb coordination by combining experimental data and computer simulation to shed light on how a population of spinal neurons may coordinate hind and fore limbs during locomotion at different speeds.

---

## [Decision Letter]

**Decision letter after peer review:**

Thank you for submitting your article "The role of V3 neurons in speed-dependent interlimb coordination during locomotion in mice" for consideration by *eLife*. Your article has been reviewed by 2 peer reviewers, and the evaluation has been overseen by a Reviewing Editor and Ronald Calabrese as the Senior Editor. The following individuals involved in review of your submission have agreed to reveal their identity: Tuan V Bui (Reviewer #1); Avihu Klar (Reviewer #2).

Essential revisions:

The present study aims at identifying a sub-population of V3 neurons that would be involved in coordinating hindlimb and forelimb movements, thus influencing locomotor speed and gait adaptation. These long-projecting V3 neurons (LPNs) would be excitatory, express the transcription factor Sim1, are located at the lumber level, are ascending with contralateral cervical projections. Their experimental silencing changes speed/gait adaptation. Experimental data can be replicated using an already existing computational model of the locomotor circuitry incremented with the newly identified V3 aLPN neurons. Despite the fact that the data presented here are addressing the role of spinal interneurons in hind/forelimb coordination, which has been only insufficiently examined so far, the study suffers from several problems that should be addressed before the paper can be judged suitable for publication in *eLife*.

The major concerns are as follows:

1) The methodological approaches are not always targeting specifically the neurons of interest. For instance, the retrograde labeling method used to label V3 aLPN neurons is not specific enough (see reviewers' comments). We recommend at least that the author amend their diagram in Figure 1A1 to show that they could also be staining en passant axons and that they state this in the text as a source of uncertainty. Also the blockade of glutamatergic signaling at the lumbar level as it is performed blocks all connections and not only those involving V3 aLPN neurons. It is right that the stimulation is specific but no direct conclusions can be reached from these experiments on the exclusive role of V3 aLPN neurons specifically. Connections of these neurons with all other ones located at the same level will also be affected. Conclusions here must be made cautiously and limitations addressed in the discussion.

2) A lack of description of the connections between the neurons of interest (V3 aLPN) and the other sub-population of V3 neurons identified in previous studies (namely the commissural V3), and also with interneurons known to be involved in the control of gaiting and locomotor speed. In the same vain, it is not clear whether the kinetics' of the recordings points to monosynaptic or polysynaptic connectivity. Therefore, it is imperative to demonstrate at least that the lumbar aV3 are either pre-cervical-MN, or innervate the pre-cervical-MN, as indicated in Figure 7. We are aware that demonstrating the anatomy of the lumbar aV3 is not straightforward due to the transient expression of Sim1 at embryonic stages and the amount of work it represents. Therefore, we are willing to comprise on obtaining the anatomy data, if more accurate electrophysiology data are obtained even if it requires the traditional "old-fashioned" electrophysiology to support the authors' model.

3) It appears incorrect to use data obtained on the V3OFF mice (in which all V3 neuronal subtypes are targeted) to conclude on the specific role of V3 aLPN neurons. For example some models can account for the phenotype of the locomotion following deletion of vGlut2 in V3 neurons. Maybe the cervical V3 neurons receive inputs from other lumbar interneurons? Hence, impairing their activity may result in similar gait impairments. We think that manipulating the lumbar aV3, or at least, the entire lumbar V3 is imperative for substantiating the model. The authors should find ways to eliminate only V3 aLPN neurons as suggested by reviewers (or at least to strongly moderate their conclusions).

4) In the present study, the computational model replicates quite well data observed in vitro. This is a powerful tool to develop new hypothesis. But the main interest is then to subsequently test at least some of these hypothesis. Here a long part of the paper is dedicated to describe the model (which is now quite complex) and to establish that it does what the modeler wants, but it would strengthen the paper if at least one of the prediction could be tested here. Additional experiments are required here, as suggested by the reviewers.

5) In a general manner, figures are very loaded and complicated to understand at a first sight. It would be useful for the readers to be guided towards what are the important panels. For example on Figures 4 and 11 could you highlight one way or another the panels on which the readers should focus on. Also data presented in Figure 1b requires details. Indeed traces are small and presented with an inappropriate time scale: the coordination between lumbar/cervical, and left/right is not visible. Could you add samples at a different time scale? And quantification of the intensity of activity before, during and after the light stimulation should be provided (see comments of reviewers).

*Reviewer #1:*

In this study, the authors sought to describe the role of a set of spinal neurons named V3 interneurons in the coordination of hindlimbs and forelimbs in locomotor control in mice.

The authors first mapped how these neurons in the lumbar section of the spinal cord, involved in the control of the hindlimbs, were connected to cervical section controlling the forelimbs. They then analyzed stimulating or removing these neurons affected locomotor output using optogenetics and transgenic silencing. Finally, the authors used computational modelling to infer possible connections between these neurons with other spinal neurons in the forelimb and hindlimb segments of the spinal cord that could explain the results that were observed during their experiments.

The major strengths of the study was the rigorous analysis of locomotor deficits in animals lacking the V3 neurons. The replication of these phenotypes in the computational models generates testable hypotheses about the connectivity of the V3 neurons with the rest of the spinal cord.

While the experimental and computational methodology is rigorous, an analysis of the phase relationship of cells, between cells controlling limb coordination involving ascending V3s would provide greater insights into the operation of the spinal locomotor circuits.

The experimental and the simulation results support the role of ascending V3 interneurons in the control of left-right coordination across forelimb and hindlimbs.

The computational models build upon previous models from the authors. They are useful tools for the community to study the operation of spinal circuits to control the many parameters of locomotor control.

Suggestions:

1. Can you go into more details regarding the rationale of the diagonal aV3 populations? I would suggest moving the section Circuit interactions mediated by V3 subpopulations before describing the model connectivity because it provides a rationale for the connectivity of these neurons in the model.

2. Please describe the rationale for modelling a descending drive to aV3s.

3. I don't fully agree that the increase in activity in the V3 population seen in the model would be reflected in the increased cFos expression, which is a proxy for the number of neurons that are activated. From the model, it looks like the population is fully active at lower speeds but their firing frequency increases with speed.

4. I agree that the model qualitatively replicates the experimental results with the V3OFF mice. In addition, can you describe where the model differs from experimental results for the V3OFF data?

5. Could you generate phase plots to show how the relationship between the activity of relevant cell populations shift at different speeds and without aV3s?

6. In the analysis of the computer simulations, how was the minimal burst amplitude determined? Were there bursts that could have just been due to noise in RG activity?

*Reviewer #2:*

The Zhang et al., study addresses the role of lumbar to cervical ascending V3 interneurons in controlling intralimb coordination. The role of spinal interneurons in control of left/right and flexor/extensor alternations has been studied extensively, while their role in hind/forelimb coordination is yet insufficiently studied. In this respect, revealing the role of lumbar-cervical V3-aLPN is essential and timely. The authors are using retrograde labeling and physiological recording to demonstrate the lumbar to cervical connectivity of V3, kinematic analysis of V3off mice in which the vGlut2 was conditionally removed in all V3 neurons, and present an updated model. Based on computationally testing their theoretical model, the authors also hypothesize the outcome of silencing only the V3-aLPN subpopulation. There are two main concerns that question the validity of the findings: 1) the data that supports the lumbar-to-brachial connectivity are not sufficiently convincing, 2) the conclusion about the role of V3 in intralimb coordination is based on the silencing of the entire V3 populations not the V3-aLPN subpopulation.

1) The data in figure 1 demonstrate the lumbar-cervical connectivity of V3 neurons. The authors are using retrograde labeling attained by cervical injection of CTB. However, CTB may also label transneuronal processes (passing by axons). Two previous papers cited extensively throughout this manuscript mapped the descending cervical-to-lumbar connectome of spinal interneurons (Ruder et al., 2016 and Flynn et al., 2017). In these papers, the researchers use more reliable retrograde labeling methods: Rabies and Fluorogold. Since the central issue of the manuscript relies on the lumbar-cervical circuit, more data should be provided. For example, 1) The distribution of V3-lumbar synapses in the cervical level. 2) Labeling the lumbar aLPN by using alternative retrograde labeling methods.

The experiment presented in 1B is supposed to provide physiological verification to the lumbar-cervical connectivity. The authors present one representative trace of rectified ENG recordings following single light activation of the lumbar V3 neurons. Quantification of the excitation level before-during-after blue light emission in several (at list 10) episodes should be included. By observing figure B1, I am not convinced that Left C8 is activated at all.

2) The paper's main conclusion is: "The proposed V3 aLPN connections support diagonal synchronization necessary for trot whereas the local V3 CIN connections support left-right synchronization necessary for gallop and bound". The theoretical model supports the following hypothesis: "selective deletion of only V3 aLPNs in the model allowed for stable coordination of limb activities at high speeds, i.e., during gallop and bound, whereas trot was completely lost and the model transitioned from walk directly to gallop and bound". The suggested experiment is imperative to prove this hypothesis, and the authors have the means to perform this experiment.

---

## [Author Response]

Essential revisions:The present study aims at identifying a sub-population of V3 neurons that would be involved in coordinating hindlimb and forelimb movements, thus influencing locomotor speed and gait adaptation. These long-projecting V3 neurons (LPNs) would be excitatory, express the transcription factor Sim1, are located at the lumber level, are ascending with contralateral cervical projections. Their experimental silencing changes speed/gait adaptation. Experimental data can be replicated using an already existing computational model of the locomotor circuitry incremented with the newly identified V3 aLPN neurons. Despite the fact that the data presented here are addressing the role of spinal interneurons in hind/forelimb coordination, which has been only insufficiently examined so far, the study suffers from several problems that should be addressed before the paper can be judged suitable for publication in eLife.

We thank all the Reviewers and Editors for the careful review, thoughtful comments, and efforts toward improving our manuscript. We have addressed all of the concerns and revised the manuscript accordingly. Our response to the essential and each Reviewer’s concerns are summarized below.

The major concerns are as follows:1) The methodological approaches are not always targeting specifically the neurons of interest. For instance, the retrograde labeling method used to label V3 aLPN neurons is not specific enough (see reviewers' comments). We recommend at least that the author amend their diagram in Figure 1A1 to show that they could also be staining en passant axons and that they state this in the text as a source of uncertainty.

We thank the Reviewers for pointing this out. We agree that CTB can be taken up by fibers of passage. We have changed the diagram in Figure 1A1 to show the potential en passant axons, as suggested. We also now explicitly acknowledge in the Discussion that some of the CTB-labelled lumbar V3 aLPN neurons may pass the cervical region (see the beginning of the third paragraph in the section: “Lumbar V3 neurons with ascending long propriospinal projections to the contralateral cervical region”).

To further demonstrate that lumbar V3 aLPNs directly innervate neurons in the contralateral cervical locomotor region, we have performed a new experiment presented now in new Figure 1B1,B2 and Figure 1 —figure supplement 1. In this experiment, we injected AAVhSyn-GFP in the lumbar region of Sim1CreTdtom mice and performed Vglut2 immunostaining. Triple-labelled (VGluT2^+^, GFP+, V3-tdTom+) terminals in the regions of motor neurons and lamina VII/VIII interneurons in cervical spinal cord indicate that lumbar V3 aLPNs make direct connections with cervical spinal motor neurons and interneurons.

In addition, to demonstrate more clearly that activation of V3 neurons in the lumbar region can directly evoke the motor output on the cervical ventral roots, we have modified Figure 1C1-C3 (former Figure 1B1-B4) and added Figure 1—figure supplements 2 and 3.

We have included corresponding descriptions in the Results (see second and third paragraphs in section: “Lumbar propriospinal V3 interneurons provide ascending excitatory drives to the contralateral cervical locomotor circuits”) and Discussion (section: Lumbar V3 neurons with ascending long propriospinal projections to the contralateral cervical region). We believe that our new anatomical data, taken together with our original anatomy and electrophysiology experiments, makes the case that the lumbar cord contains a subset of V3 neurons which project directly to the contralateral cervical region.

Also the blockade of glutamatergic signaling at the lumbar level as it is performed blocks all connections and not only those involving V3 aLPN neurons. It is right that the stimulation is specific but no direct conclusions can be reached from these experiments on the exclusive role of V3 aLPN neurons specifically. Connections of these neurons with all other ones located at the same level will also be affected. Conclusions here must be made cautiously and limitations addressed in the discussion.

It is true that the glutamatergic signaling blockade is not specific and that selective activation or silencing of V3 aLPNs would be an ideal experiment to reveal their specific function. Unfortunately, we do not currently have experimental tools for specific targeting of V3 aLPNs exclusively. Our data showed that photoactivation of all lumbar V3s resulted in a motor response in the cervical cord. Therefore, we suggested that this response was provided by projections of V3 aLPN neurons. It could also be that light activated some other (not aLPN) lumbar V3 neurons that in turn activated some other (not V3) lumbar aLPNs projecting to the cervical cord. However, we also showed the effect of lumbar V3 stimulation on the activity of cervical roots maintained after adding the KA that blocked all glutamatergic synapses in the lumbar cord, which eliminated the above possibility. Taking together with our anatomy results, these experiments support the conclusion that the lumbar cord contains at least a subpopulation of aLPNs, whose activation leads to a motor response in the cervical cord. This is now discussed in the Discussion (fourth paragraph in section: “Lumbar V3 neurons with ascending long propriospinal projections to the contralateral cervical region“).

2) A lack of description of the connections between the neurons of interest (V3 aLPN) and the other sub-population of V3 neurons identified in previous studies (namely the commissural V3), and also with interneurons known to be involved in the control of gaiting and locomotor speed. In the same vain, it is not clear whether the kinetics' of the recordings points to monosynaptic or polysynaptic connectivity. Therefore, it is imperative to demonstrate at least that the lumbar aV3 are either pre-cervical-MN, or innervate the pre-cervical-MN, as indicated in Figure 7. We are aware that demonstrating the anatomy of the lumbar aV3 is not straightforward due to the transient expression of Sim1 at embryonic stages and the amount of work it represents. Therefore, we are willing to comprise on obtaining the anatomy data, if more accurate electrophysiology data are obtained even if it requires the traditional "old-fashioned" electrophysiology to support the authors' model.

Unfortunately, connections of V3 aLPN neurons with other identified interneurons (including local V3 interneurons) are currently mainly unknown and it would be too ambitious to hope that they could be revealed in this study. We have mentioned this in the Discussion (fourth paragraph in section: “Lumbar V3 neurons with ascending long propriospinal projections to the contralateral cervical region“). We could only specify lumbar V3 aLPN synapses on specific identified motoneurons and interneurons of the cervical locomotor region, such as in laminae VII and IX, using injection of AAV2/9-hSyn-eGFP in the lumbar spinal cords of Sim1tdTom mice (see new Figure 1B1,B2 and Figure1 —figure supplement 1). This is now described in the Discussion (last paragraph of section: “Lumbar propriospinal V3 interneurons provide ascending excitatory drives to the contralateral cervical locomotor circuits”).

Moreover, because of the current insufficiency of existing experimental data on connectivity, most of neuronal connections in the spinal network, including the connections of local and long propriospinal V3 interneurons, were only suggested in our model. At the same time, we hope that the ability of the model to reproduce a large body of experimental results (our and others’) provides indirect validation to the connectivity proposed in the model, and hence the proposed connectivity represent modeling predictions for future experimental studies. This is now discussed in the Discussion (first paragraph of the section: “Modeling predictions and limitations of the model”).

Classical electrophysiological methods to evaluate if the mono-, di-, or oligo-synaptic connections of V3 aLPNs to cervical motoneurons exist would be difficult to interpret. The lack of myelination in neonatal preparations and room temperature recordings would induce a substantial amount of variability in the conduction velocity and latency. Our new experiments (new Figure 1B1,B2 and Figure 1 —figure supplement 1) demonstrate that V3 aLPNs broadly project to the cervical gray matter regions and innervate there both interneurons and motoneurons (also please see our response to point 1 above).

It should be noted that the present computational model does not include motoneurons, but only simulates pre-motor rhythm-generating, commissural (left-right) and propriospinal (lumbar-cervical) coordinating circuits. Therefore, connections to motoneurons were not directly considered in the model.

3) It appears incorrect to use data obtained on the V3OFF mice (in which all V3 neuronal subtypes are targeted) to conclude on the specific role of V3 aLPN neurons. For example some models can account for the phenotype of the locomotion following deletion of vGlut2 in V3 neurons. Maybe the cervical V3 neurons receive inputs from other lumbar interneurons? Hence, impairing their activity may result in similar gait impairments. We think that manipulating the lumbar aV3, or at least, the entire lumbar V3 is imperative for substantiating the model. The authors should find ways to eliminate only V3 aLPN neurons as suggested by reviewers (or at least to strongly moderate their conclusions).

We agree with the Reviewers that the phenotypes shown in V3^OFF^ mice are the collective effects of the entire V3 populations. However, as we mentioned above in our response to point 1 (and now in the paper), we do not currently have experimental tools for specific targeting of V3 aLPNs exclusively. As the Reviewers noted above, Sim1 is not expressed beyond embryonic stages, which means there is no expression of cre in postnatal spinal cord. Therefore, we cannot directly target V3 aLPNs using retrograde transporting-virus carrying DTR or DREADD, that is dually controlled by cre and flpO injected in cervical region and local expression AAV-flpO injected in the lumbar, as is possible when cre expression persists. There is a potential alternative, using GFP dependent expression of cre (Cre-DOG) (Tang et al., 2015), but its efficiency is still questionable, and it will require the intersect of expression of four viruses (2 for Cre-DOG, a FlpO, and the cre/flp-dependent DTR or DREADD). This is likely to take years to make it work, which is far beyond the scope of this work. We have further adjusted our text in the paper to clarify what conclusions can be drawn from the experimental data at hand, and which ones are predictions/suggestions from the computational model.

Since we were unable to specifically manipulate the V3 aLPNs, we have combined modeling and experimental approaches in our study to predict functional consequences of eliminating individual subpopulations in our model, which could be tested by future experiments. Our model incorporated the lumbar V3 neurons with ascending projections to the contralateral cervical circuits. This, however, as pointed out by the Reviewers, does not necessarily exclude the existence of other (non-V3) lumbar neurons that homolaterally excite some commissural cervical V3 neurons, which in turn directly activate contralateral cervical RG (flexor half-center). Indeed, in this case, the deletion of Vglut2 in all V3 would have a similar effect. Yet, we have not considered such a possibility in the model because for two reasons. First, when we optogenetically activated all lumbar V3 neurons in vitro, we recorded an increase of activity in the cervical C5 and C8 roots, which was not eliminated by suppression glutamatergic transmission within the lumbar cord (Figure 1C1-C3 and Figure 1 —figure supplement 3). This provided strong support of lumbar V3 neurons projecting directly to cervical cord, which we include in our model. Second, the ascending pathways proposed by the Reviewers do not require a presence of ascending projections from the lumbar V3 cells to the contralateral cervical region, which we found in our experimental studies (see Figure 1A1-A4, and new Figure 1B1,B2 and Figure 1-supplemetal figure 1). Thus, there is concrete evidence in support of the connectivity suggested by our model, but not for the alternative suggested by the Reviewer. We now discuss this issue in the Discussion (third paragraph of section: “Lumbar propriospinal V3 interneurons provide ascending excitatory drives to the contralateral cervical locomotor circuits”) and clearly state that we cannot exclude the existence of additional ascending pathways, not involving V3 aLPN neurons.

4) In the present study, the computational model replicates quite well data observed in vitro. This is a powerful tool to develop new hypothesis. But the main interest is then to subsequently test at least some of these hypothesis. Here a long part of the paper is dedicated to describe the model (which is now quite complex) and to establish that it does what the modeler wants, but it would strengthen the paper if at least one of the prediction could be tested here. Additional experiments are required here, as suggested by the reviewers.

We fully agree with the Reviewers that the paper would be stronger if it would include experimental testing of modeling predictions. Since most of the architectural/structural data, including known types of neurons and their connectivity are unknown, the computational model presented here relies on multiple assumptions, which were specifically made to reproduce a series of known behaviors. Thus, these suggested elements (neuron types) and connectivity represent the modeling predictions, which should then be tested to validate the model. Yet, in case of the mammalian spinal circuitry, testing of the suggested neuronal connectome represents a very difficult task that requires manipulation of identified neuron types located in different parts of the spinal cord. Each of these experiments would require substantial time and effort or might even be infeasible with current methodologies.

For example, the main prediction of our model here is the presence of different types of V3 neuron populations: the local V3 CINs, operating within lumbar and cervical cords, and the propriospinal V3s with ascending projections from lumbar to the contralateral cervical regions V3 aLPNs. The model was able to reproduce the effects of silencing all V3 neurons.

It was also used to predict the effects of selective silencing or removal of V3 aLPNs (Figure 11 and its supplement 1). However, as we stated above, we currently do not have experimental tools to test this prediction, but such tools or methods can be developed in the future. While the present study was mainly focused on V3 neurons, the importance of the proposed model would be low if the model could only reproduce data regarding the manipulation of V3 neurons described here. Our ultimate goal has always been to develop a generalized model that would be consistent with a large body of data obtained in multiple laboratories under different experimental conditions. Major parts of the network architecture used in the present model were based on our previous models (Rybak et al., 2015; Shevtsova et al., 2015; Danner et al., 2016, 2017, 2019; Ausborn et al., 2019), which are able to reproduce experimental data from multiple studies using selective removal, silencing, inhibition, or activation of different types of genetically identified spinal interneurons. Using these previous models as a basis, we were very careful to make sure that the updated model presented here is still able to reproduce the previous experimental data. Indeed, it is able to reproduce the effects of genetic ablation of V0_V_, V0_D_ and all V0 commissural interneurons (Talpalar et al., 2013; Bellardita and Kiehn 2015), inactivation of descending long propriospinal neurons (Ruder et al., 2016), and uni- and bilateral stimulations of V3 neurons (Danner et al., 2019). Therefore, despite the relative complexity of the model, the number of different experimental phenomena that the final model can reproduce provides additional validation of the proposed model. This is now discussed in the Discussion (first two paragraphs of section: “Modeling predictions and limitations of the model”).

5) In a general manner, figures are very loaded and complicated to understand at a first sight. It would be useful for the readers to be guided towards what are the important panels. For example on Figures 4 and 11 could you highlight one way or another the panels on which the readers should focus on. Also data presented in Figure 1b requires details. Indeed traces are small and presented with an inappropriate time scale: the coordination between lumbar/cervical, and left/right is not visible. Could you add samples at a different time scale? And quantification of the intensity of activity before, during and after the light stimulation should be provided (see comments of reviewers).

Old Figures 4 and 11 (current Figures 5 and 12, respectively) and old Figure 1B1-B4

(current Figure 1C1-C3) have been modified to address the Reviewers’ concern.

In current Figure 1C2, we pooled 12 individual episodes under each condition: before and during kynurenic acid application and following washout. The recordings were rectified, averaged, and smoothed to show the overall activity for each root. The unprocessed, raw data is now shown in Figure 1 —figure supplement 3. We also overlapped the traces for three conditions and increased the vertical scale in this panel (Figure 1C2) to make the effect of light application more obvious. However, we do not see the necessity of increasing the time scale in this figure because these experiments were performed in non-locomotor conditions in which the coordination between activities in lumbar/cervical and left/right roots was not expected. We also added a new panel to this figure (Figure 1C3) that shows quantification of the intensity of the response to light stimulation before and during KA application and following washout.

Reviewer #1:Suggestions:1. Can you go into more details regarding the rationale of the diagonal aV3 populations? I would suggest moving the section Circuit interactions mediated by V3 subpopulations before describing the model connectivity because it provides a rationale for the connectivity of these neurons in the model.

Incorporating the diagonal aV3 populations in the model was based on our experimental studies that showed projection of lumbar V3 aLPN to the contralateral cervical region (Figure 1A1-A4,B1,B2 and Figure 1—figure supplement 1) that were described in the very beginning of the Results before the description of the model. We believe that the section “Circuit interactions mediated by V3 subpopulations” requires knowledge of the other major circuit elements and we prefer to keep the current sequence of model description.

2. Please describe the rationale for modelling a descending drive to aV3s.

The descending drive defines locomotor frequency and speed in the model. Drive to aV3 populations was included in the model to provide gradual recruitment of aV3 neurons and increase of their firing frequency with increasing speed supporting switching from walk to trot at the appropriate speed.

3. I don't fully agree that the increase in activity in the V3 population seen in the model would be reflected in the increased cFos expression, which is a proxy for the number of neurons that are activated. From the model, it looks like the population is fully active at lower speeds but their firing frequency increases with speed.

We thank the Reviewer for this comment and replaced this figure (old Figure 9) with the new Figure 10 showing another example of simulation in which some of aV3 neurons were not active at low speed and were recruited with increasing speed. In the model, this effect and the simultaneous increase in firing rate of aV3 neurons support the transition between walk and trot.

4. I agree that the model qualitatively replicates the experimental results with the V3OFF mice. In addition, can you describe where the model differs from experimental results for the V3OFF data?

The model was developed to reproduce the presented experimental data and we did our best to achieve this. Nevertheless, our model (as any other of the existing models) has a number of limitations. These limitations and potential disagreements with the real biological system are now described in the Discussion (see section:“Modeling predictions and limitations of the model”).

5. Could you generate phase plots to show how the relationship between the activity of relevant cell populations shift at different speeds and without aV3s?

We thank the Reviewer for this suggestion. We generated circular phase plots showing phase differences between activities of the reference (left hind E) populations and other RG (rh-E, lf-E, rf-E, lh-F, lf-F, rh-F, and rF-F) and the left V3 (lh-V3-E, lh-V3-F, and lh-aV3) populations for the intact model and after deletion of all V3 or aV3 neurons. The phase diagrams are shown for 3 values of α (corresponding to three different speeds) and are now included as a new figure (Figure 11 —figure supplement 1).

6. In the analysis of the computer simulations, how was the minimal burst amplitude determined? Were there bursts that could have just been due to noise in RG activity?

We thank the Reviewer for this comment. The threshold to calculate bust onsets was determined generally as 10% of difference between the maximal and minimal values of integrated activity of a particular RG population. In most cases, this threshold value was sufficient to eliminate low amplitude bursts that appeared due to noise in RG activity. To build the new figure (Figure 11 —figure supplement 1), the threshold was increased to 30% of the difference for V3 population because of a higher level of background activity in these populations. We now mention this in Materials and methods.

Reviewer #2:The Zhang et al., study addresses the role of lumbar to cervical ascending V3 interneurons in controlling intralimb coordination. The role of spinal interneurons in control of left/right and flexor/extensor alternations has been studied extensively, while their role in hind/forelimb coordination is yet insufficiently studied. In this respect, revealing the role of lumbar-cervical V3-aLPN is essential and timely. The authors are using retrograde labeling and physiological recording to demonstrate the lumbar to cervical connectivity of V3, kinematic analysis of V3off mice in which the vGlut2 was conditionally removed in all V3 neurons, and present an updated model. Based on computationally testing their theoretical model, the authors also hypothesize the outcome of silencing only the V3-aLPN subpopulation. There are two main concerns that question the validity of the findings: 1) the data that supports the lumbar-to-brachial connectivity are not sufficiently convincing, 2) the conclusion about the role of V3 in intralimb coordination is based on the silencing of the entire V3 populations not the V3-aLPN subpopulation.

We thank the Reviewer #2 for these comments and address the two main concerns below.

1) The data in figure 1 demonstrate the lumbar-cervical connectivity of V3 neurons. The authors are using retrograde labeling attained by cervical injection of CTB. However, CTB may also label transneuronal processes (passing by axons). Two previous papers cited extensively throughout this manuscript mapped the descending cervical-to-lumbar connectome of spinal interneurons (Ruder et al., 2016 and Flynn et al., 2017). In these papers, the researchers use more reliable retrograde labeling methods: Rabies and Fluorogold. Since the central issue of the manuscript relies on the lumbar-cervical circuit, more data should be provided. For example, 1) The distribution of V3-lumbar synapses in the cervical level. 2) Labeling the lumbar aLPN by using alternative retrograde labeling methods.

We responded to this concern above in our response to the major concerns. We agree that CTB can be taken up by fibers of passage and show this now in the updated Figure 1A. However, based on our experience, the same is true for FG, which has also been reported by others (Dado et al., 1990; Chen and Aston-Jones, 1995; Brown et al., 2021). Therefore, as the Reviewer recommended, we have added anterograde tracing experiment using AAVhSyn-GFP injected into the lumbar cord (Figure 1B1,B2 and Figure 1 —figure supplement 1). These figures show glutamatergic synaptic terminals (Vglut2 staining) in cervical spinal cord that originate from lumbar (GFP), V3 (tdTom) neurons. This issue is now described in the Results (see second and third paragraphs in section: “Lumbar propriospinal V3 interneurons provide ascending excitatory drives to the contralateral cervical locomotor circuits”) and Discussion (section: Lumbar V3 neurons with ascending long propriospinal projections to the contralateral cervical region).

The experiment presented in 1B is supposed to provide physiological verification to the lumbar-cervical connectivity. The authors present one representative trace of rectified ENG recordings following single light activation of the lumbar V3 neurons. Quantification of the excitation level before-during-after blue light emission in several (at list 10) episodes should be included. By observing figure B1, I am not convinced that Left C8 is activated at all.

We agree with the Reviewer that presentation of our data in old Figure 1B1-B4 was too busy and that some traces were small to be seen well. This figure has been replaced with new Figure 1C1-C3 and two figure supplements were added for clarity. Specifically, new Figure 1C2 shows overall activity of each root that averaged over 12 episodes for all three conditions (Ringer, KA, and Washout). Raw recordings for all three conditions are shown in Figure 1 —figure supplement 3. Based on individual traces, we measured the response in all roots to light stimulation in the same three conditions (Figure 1C3). We also added a proof of principle experiment (see Figure 1 —figure supplement 2), which shows that photoactivation of V3 neurons in the cervical cord only evoked motor output in cervical ventral roots, but not in lumbar roots. In contrast, photoactivation of lumbar V3 neurons generated bursts in both lumbar and cervical ventral roots. Furthermore, our results clearly show that the lumbar photoactivation of V3 neurons consistently evoked cervical motor output, even when local excitatory synapses were blocked (Figure 1 —figure supplement 2).

2) The paper's main conclusion is: "The proposed V3 aLPN connections support diagonal synchronization necessary for trot whereas the local V3 CIN connections support left-right synchronization necessary for gallop and bound". The theoretical model supports the following hypothesis: "selective deletion of only V3 aLPNs in the model allowed for stable coordination of limb activities at high speeds, i.e., during gallop and bound, whereas trot was completely lost and the model transitioned from walk directly to gallop and bound". The suggested experiment is imperative to prove this hypothesis, and the authors have the means to perform this experiment.

Please see the response to main concern 3. Unfortunately, as detailed above, we are unable to selectively target this subpopulation experimentally to prove our prediction of this study, but this further emphasizes the importance of our modelling work. This issue is now mentioned and/or discussed in several places of the paper.

References

Ausborn J, Shevtsova NA, Caggiano V, Danner SM, Rybak IA. 2019. Computational modeling of brainstem circuits controlling locomotor frequency and gait. *eLife* 8:e43587. doi:10.7554/*eLife*.43587

Bellardita C, Kiehn O. 2015. Phenotypic characterization of speed-associated gait changes in mice reveals modular organization of locomotor networks. Curr Biol 25:1426–1436. doi:10.1016/j.cub.2015.04.005

Brown BL, Zalla RM, Shepard CT, Howard RM, Kopechek JA, Magnuson DSK, Whittemore SR. 2021. Dual-viral transduction utilizing highly efficient retrograde lentivirus improves labeling of long propriospinal neurons. Front Neuroanat 15:635921. doi:10.3389/fnana.2021.635921

Chen S, Aston-Jones G. 1995. Evidence that cholera toxin B subunit (CTb) can be avidly taken up and transported by fibers of passage. Brain Res 674(1):107-11. doi:10.1016/00068993(95)00020-q

Chopek JW, Nascimento F, Beato M, Brownstone RM, Zhang Y. 2018. Sub-populations of spinal V3 interneurons form focal modules of layered pre-motor microcircuits. Cell Rep 25:146– 156. doi:10.1016/j.celrep.2018.08.095

Dado RJ, Burstein R, Cliffer KD, Giesler GJ Jr. 1990. Evidence that Fluoro-Gold can be transported avidly through fibers of passage. Brain Res 533(2):329-33. doi:10.1016/00068993(90)91358-n

Danner SM, Shevtsova NA, Frigon A, Rybak IA. 2017. Computational modeling of spinal circuits controlling limb coordination and gaits in quadrupeds. *eLife* 6:e31050. doi:10.7554/*eLife*.31050

Danner SM, Wilshin SD, Shevtsova NA, Rybak IA. 2016. Central control of interlimb coordination and speed-dependent gait expression in quadrupeds. J Physiol 594:6947–6967. doi:10.1113/JP272787

Danner SM, Zhang H, Shevtsova NA, Borowska-Fielding J, Deska-Gauthier D, Rybak IA, Zhang Y. 2019. Spinal V3 interneurons and left–right coordination in mammalian locomotion.

Front Cell Neurosci 13:516. doi:10.3389/fncel.2019.00516

Flynn JR, Conn VL, Boyle KA, Hughes DI, Watanabe M, Velasquez T, Goulding MD, Callister RJ, Graham BA. 2017. Anatomical and molecular properties of long descending propriospinal neurons in mice. Front Neuroanat 11. doi:10.3389/fnana.2017.00005

Ruder L, Takeoka A, Arber S. 2016. Long-distance descending spinal neurons ensure quadrupedal locomotor stability. Neuron 92:1063–1078. doi:10.1016/j.neuron.2016.10.032

Rybak IA, Dougherty KJ, Shevtsova NA. 2015. Organization of the mammalian locomotor CPG: Review of computational model and circuit architectures based on genetically identified spinal interneurons. ENeuro 2:ENEURO.0069-15.2015. doi:10.1523/ENEURO.006915.2015

Shevtsova NA, Talpalar AE, Markin SN, Harris-Warrick RM, Kiehn O, Rybak IA. 2015. Organization of left-right coordination of neuronal activity in the mammalian spinal cord: Insights from computational modelling. J Physiol 593:2403–2426. doi:10.1113/JP270121

Talpalar AE, Bouvier J, Borgius L, Fortin G, Pierani A, Kiehn O. 2013. Dual-mode operation of neuronal networks involved in left–right alternation. Nature 500:85–88. doi:10.1038/nature12286

Tang JC, Rudolph S, Dhande OS, Abraira VE, Choi S, Lapan SW, Drew IR, Drokhlyansky E, Huberman AD, Regehr WG, Cepko CL. 2015. Cell type-specific manipulation with GFPdependent Cre recombinase. Nat Neurosci 8(9):1334-41. doi:10.1038/nn.4081

Zhang Y, Narayan S, Geiman E, Lanuza GM, Velasquez T, Shanks B, Akay T, Dyck J, Pearson K, Gosgnach S, Fan CM, Goulding M. 2008. V3 spinal neurons establish a robust and balanced locomotor rhythm during walking. Neuron 60:84–96. doi:10.1016/j.neuron.2008.09.027